# Parametric Scaling Law of Tuning Bias in Conformal Prediction

Hao Zeng [* 1]   Kangdao Liu [* 1 2]   Bingyi Jing [1]   Hongxin Wei [1]

## Abstract

Conformal prediction is a popular framework of uncertainty quantification that constructs prediction sets with coverage guarantees. To uphold the exchangeability assumption, many conformal prediction methods necessitate an additional hold-out set for parameter tuning. Yet, the impact of violating this principle on coverage remains underexplored, making it ambiguous in practical applications. In this work, we empirically find that the tuning bias - the coverage gap introduced by leveraging the same dataset for tuning and calibration, is negligible for simple parameter tuning in many conformal prediction methods. In particular, we observe the scaling law of the tuning bias: this bias increases with parameter space complexity and decreases with calibration set size. Formally, we establish a theoretical framework to quantify the tuning bias and provide rigorous proof for the scaling law of the tuning bias by deriving its upper bound. In the end, we discuss how to reduce the tuning bias, guided by the theories we developed.

## 1. Introduction

Quantifying uncertainty in predictions is crucial for the safe deployment of machine learning, particularly in high-risk domains such as financial decision-making and medical diagnostics. Conformal prediction stands out as a promising statistical framework for quantifying uncertainty in the predictions of any predictive algorithm (Papadopoulos, 2008; Vovk, 2012; Vovk et al., 2005). It transforms point predictions into prediction sets guaranteed to contain ground truths with a user-specified coverage rate. Under the assumption of data exchangeability, these prediction sets offer non-asymptotic coverage guarantees without distributional assumptions. However, the assumption may not always hold

due to the inherent characteristics of the data or its usage in practice, such as time series analysis, domain shifts, and parameter tuning (Barber et al., 2023; Oliveira et al., 2024). Therefore, it is essential to understand and quantify the coverage gap that arises from the violation of exchangeability.

Many conformal prediction methods utilize a hold-out set for *parameter tuning*, in addition to the calibration set. For example, (1) RAPS (Angelopoulos et al., 2021) and SAPS (Huang et al., 2024) search their parameters on the hold-out set; (2) score aggregation methods (Fan et al., 2024; Luo & Zhou, 2024; Yang & Kuchibhotla, 2024) optimize the selection or weights for candidate scores; (3) confidence calibration methods (Dabah & Tirer, 2024; Guo et al., 2017; Xi et al., 2024) tune the scale parameters for better calibration; (4) training methods (Liu et al., 2025; Stutz et al., 2022) fine-tune models on the hold-out set. Previous works claim that leveraging the same hold-out set for tuning and calibration will destroy the exchangeability (Angelopoulos et al., 2021; Dabah & Tirer, 2024; Xi et al., 2024; Yang & Kuchibhotla, 2024), but they did not provide a formal quantification on the coverage gap introduced by the violation of exchangeability systematically. This prompts us to investigate the influence of parameter tuning on the coverage gap in the absence of a hold-out set.

In this work, we reveal a previously unrecognized phenomenon: the tuning bias - the coverage gap introduced by using the same dataset for tuning and calibration, is negligible for simple parameter tuning in conformal prediction methods. Empirically, we find that most methods maintain their coverage rates, except for vector scaling (Guo et al., 2017) and the fine-tuning version of ConfTr (Stutz et al., 2022). Furthermore, we observe the *parametric scaling law* of the tuning bias: this bias increases with parameter space complexity and decreases with calibration set size. An intuitive explanation for this phenomenon is that models with more complex parameter space require more data to tune for the optimal parameters.[1]

Formally, we propose a theoretical framework to quantify the tuning bias, which is formulated as a specifically designed constrained ERM problem (Bai et al., 2022). In particular, we measure the tuning bias using the classical

---
[*]Equal contribution [1]Department of Statistics and Data Science, Southern University of Science and Technology [2]Department of Computer and Information Science, University of Macau . Correspondence to: Hongxin Wei <weihx@sustech.edu.cn>.

*Proceedings of the 42$^{nd}$ International Conference on Machine Learning*, Vancouver, Canada. PMLR 267, 2025. Copyright 2025 by the author(s).

---
[1]Code: `https://github.com/ml-stat-Sustech/Parametric-Scaling-Law-CP-Tuning`.

empirical process within the extended parameter space (van der Vaart & Wellner, 1996). Furthermore, we derive the upper bounds of tuning bias in the cases of finite and infinite parameter spaces, respectively. We then provide rigorous proof for the parametric scaling law of tuning bias through the upper bounds, which aligns with empirical observations. Finally, we present the tuning biases of parameter tuning in several conformal prediction methods, such as RAPS, score aggregation, and confidence calibration methods.

We further discuss the potential solutions to mitigate tuning bias. With the parametric scaling law, we could increase the size of the calibration set or reduce the complexity of the parameter space to reduce the tuning bias. The former may not be practical due to data scarcity. On the latter, we discuss two cases: one is to reduce the number of parameters to tune; the other is to apply regularization on the parameter space. Our main contributions are summarized as follows:

- We identify a new phenomenon: simple parameter tuning in many conformal prediction methods leads to negligible tuning bias, where we leverage the same dataset for tuning and calibration.

- We present the *parametric scaling law* of tuning bias: this bias increases with parameter space complexity and decreases with calibration set size.

- We establish a theoretical framework to quantify tuning bias and rigorously prove the parametric scaling law of tuning bias via its upper bounds.

- We discuss the potential application of our theoretical results and explain order-preserving regularization as a possible solution to reduce tuning bias.

## 2. Background

**Conformal prediction** Conformal prediction (Vovk et al., 2005) aims to produce prediction sets that contain ground-truth labels with a desired coverage rate. Let $\mathcal{X} \subset \mathbb{R}^d$ denote the input space and $\mathcal{Y}$ denote the label space. Formally, the goal of conformal prediction is to construct a set-valued mapping $\mathcal{C} : \mathcal{X} \to 2^{\mathcal{Y}}$ that satisfies the marginal coverage:

$$\mathbb{P}(y \in \mathcal{C}(\boldsymbol{x})) \geq 1 - \alpha, \tag{1}$$

for a user-specified error rate $\alpha \in (0, 1)$, where input $\boldsymbol{x} \in \mathcal{X}$, and output $y \in \mathcal{Y}$. As a widely used procedure, split conformal prediction (Papadopoulos, 2008) initiates with a calibration step. For each sample $(\boldsymbol{x}_i, y_i)$ from the calibration set $\mathcal{D}_{\text{cal}} := \{(\boldsymbol{x}_i, y_i)\}_{i=1}^n$, we compute the *non-conformity score* $s_i := S(\boldsymbol{x}_i, y_i)$ for a score function $S : \mathcal{X} \times \mathcal{Y} \to \mathbb{R}$. The non-conformity score function $S$ measures the strangeness of a given sample. Based on the scores computed on the calibration set, we search for a threshold

$\hat{t}$ such that the probability of observing scores on test samples below the threshold satisfies a pre-specified error rate $\alpha$. Specifically, we determine the threshold by finding the $(1 - \alpha)$-th empirical quantile of the calibration scores:

$$\hat{t} = \inf \left\{ s : |\{i : S(\boldsymbol{x}_i, y_i) \leq s\}| \geq \lceil (n+1)(1-\alpha) \rceil \right\}. \tag{2}$$

For the test step with a given feature of test sample $\boldsymbol{x}$, the non-conformity score is computed for each label $y \in \mathcal{Y}$. The prediction set $\mathcal{C}(\boldsymbol{x})$ is constructed by including all labels whose non-conformity scores are below the threshold $\hat{t}$:

$$\mathcal{C}(\boldsymbol{x}) := \widehat{\mathcal{C}}_{\hat{t}}(\boldsymbol{x}) = \left\{ y \in \mathcal{Y} : S(\boldsymbol{x}, y) \leq \hat{t} \right\}, \tag{3}$$

where the notation $\widehat{\cdot}$ is used to emphasize its dependence on the calibration set $\mathcal{D}_{\text{cal}}$. Here we consider a little weaker version of independence:

**Definition 2.1** (Exchangeability). The samples from $\mathcal{D}$ are exchangeable if and only if the joint distribution of these samples is invariant under any permutation of the indices of the samples in $\mathcal{D}$.

With the assumption of exchangeability, the conformal prediction set $\mathcal{C}(\boldsymbol{x})$ in Eq. (3) has a finite-sample coverage guarantee:

**Theorem 2.2.** *(Barber et al., 2023; Lei et al., 2018) If the samples from $\mathcal{D}_{cal}$ and the test sample $(\boldsymbol{x}, y)$ are exchangeable, the conformal prediction set defined in Eq. (3) satisfies*

$$1 - \alpha \leq \mathbb{P}\left(y \in \mathcal{C}(\boldsymbol{x})\right) \leq 1 - \alpha + \frac{1}{n+1}.$$

If the assumption of exchangeability does not hold, there could be a large deviation in the coverage rate. We could use the coverage gap to measure the conformal predictor $\mathcal{C}$ for a given error rate $\alpha$ and test sample $(\boldsymbol{x}, y)$:

$$\text{CovGap}(\mathcal{C}) := |(1 - \alpha) - \mathbb{P}(y \in \mathcal{C}(\boldsymbol{x}))|.$$

**Parameter tuning** Parameter tuning is a common practice in conformal prediction. We characterize the *parameter tuning* as follows. Given a score function $S$, we can transform it using a function $\sigma$ to obtain $S^{\sigma} := \sigma \circ S \in \{ f : \mathcal{X} \times \mathcal{Y} \to \mathbb{R} \}$. When this transformation $\sigma$ is parameterized by $\lambda \in \Lambda$, where $\Lambda$ is the parameter space, we denote the transformed score as $S^{\lambda}$ for simplicity. There are some examples:

- RAPS (Angelopoulos et al., 2021) uses a scalar $\lambda$ and an integral to tune the best score function on a hold-out set, and so does SAPS (Huang et al., 2024).

- For selection and aggregation of scores problems (Fan et al., 2024; Ge et al., 2024; Luo & Zhou, 2024; Yang & Kuchibhotla, 2024), the optimal score function or weight vector is obtained from a finite number of candidates. The selection or weights could be viewed as the transformation parameter $\lambda$ to tune.

- Conformal prediction with confidence calibration (Dabah & Tirer, 2024; Guo et al., 2017; Xi et al., 2024) uses a single positive scalar $\lambda > 0$ or a vector $\lambda \in \mathbb{R}^K$ to calibrate deep learning classifiers.

- For training models from scratch in conformal prediction, they could be modified as a post-training method. Researchers fine-tune the fully connected layer or an appended layer, and the efficiency of conformal predictors could be enhanced (Huang et al., 2023; Liu et al., 2025; Stutz et al., 2022). The parameters in the layer could be viewed as the transformation parameter.

These practices tune their parameters using a hold-out set that is separate from the one used for conformal prediction calibration. Given the transformation parameter $\lambda$, we can calculate the threshold $\hat{t}$ in Eq. (2) with parameter $\lambda$. If we use the same set for both parameter tuning and conformal prediction calibration, it will destroy the data exchangeability assumption and probably introduce an additional coverage gap. We defined the additional coverage gap as:

**Definition 2.3** (Tuning Bias). For the conformal prediction with parameter tuning, $\mathcal{C}$, the tuning bias for parameter tuning on the same set as the calibration set is defined as the additional coverage gap caused by the practice, i.e.,

$$\text{TuningBias}(\mathcal{C}) = \text{CovGap}(\mathcal{C}) - \text{CovGap}(\mathcal{C}_{\text{hold-out}}),$$

where $\mathcal{C}$ is the conformal predictor with parameter tuned on the same set as the calibration set, and $\mathcal{C}_{\text{hold-out}}$ is the conformal predictor with parameter tuning using a hold-out set separated from the set used for conformal prediction.

*Remark* 2.4. Here, we clarify that understanding the tuning bias is crucial in conformal prediction practice:

- For **data-scarce scenarios**, splitting the calibration dataset is impractical in data-scarce scenarios like rare diseases, natural disaster prediction, and privacy-constrained personal data. With limited data, using separated datasets will reduce points for parameter tuning and conformal calibration, compromising the approach's effectiveness and stability. Thus, it's valuable to assess when splitting is needed, or data reuse is permissible rather than sticking to traditional practices.

- For **simple implementation**, even with sufficient data, maintaining separate sets can increase the pipeline complexity. Understanding when this separation is unnecessary—such as when tuning bias is negligible—enables simpler, more streamlined workflows while preserving coverage guarantees, offering practical relevance.

- For **foundational understanding**, exploring the tuning bias can provide an in-depth understanding of the exchangeability assumption in conformal prediction.

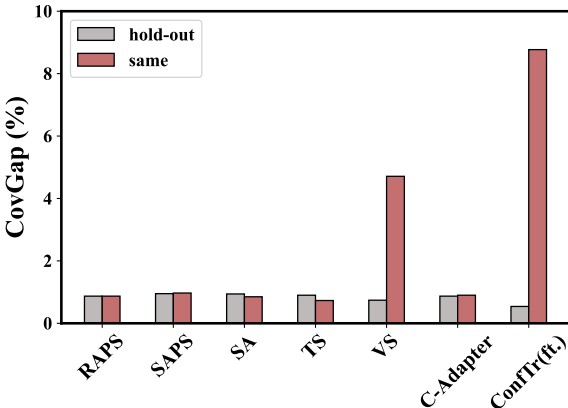

*Figure 1.* **Tuning biases of various methods in conformal prediction**, using ResNet-18 on CIFAR-100 at $\alpha = 0.1$. The APS is used except for RAPS, SAPS and SA, with a calibration set size of 1000. The "hold-out" and "same" denote that we use separate/same datasets for parameter tuning and calibration, respectively. *Tuning bias* is the difference between the coverage gaps of models tuned in the setting of *hold-out* and *same*.

In particular, the insight in this work may inspire future works in non-exchangeable conformal prediction.

It could also answer the concern, "*why not split the dataset*", for the same reasons. Indeed, it is easy to split the validation set when the provided data is sufficient. However, it can be particularly important to consider data reusing in data-scarce scenarios, like rare diseases, natural disaster prediction, and privacy-constrained personal data. With limited data, separating the dataset will further exacerbate the data scarcity problem, compromising the effectiveness of both conformal calibration and parameter tuning. In addition, exploring the tuning bias could provide an in-depth understanding of the exchangeability assumption, which may inspire future works in non-exchangeable CP.

In the next section, we first conduct an extensive empirical study of various conformal prediction methods with parameter tuning. Then, we explore how the complexity of parameter space and the size of the calibration set influence tuning bias, respectively. We empirically demonstrate the scaling laws of tuning bias on the complexity of parameter space and the size of the calibration set.

## 3. Empirical Study

### 3.1. Tuning Bias of Current Methods

We empirically investigate the potential influences of the tuning bias. The methods considered here include RAPS (Angelopoulos et al., 2021), SAPS (Huang et al., 2024), score aggregation (Luo & Zhou, 2024, SA), temperature scaling

(TS), vector scaling (VS) (Dabah & Tirer, 2024; Guo et al., 2017; Xi et al., 2024), C-Adapter (Liu et al., 2025), and the fine-tuning version of ConfTr (Stutz et al., 2022). We conduct experiments on the CIFAR-100 dataset (Krizhevsky, 2009) and use pre-trained model ResNet-18 (He et al., 2016). The introduction of methods and detailed experimental setup are provided in Appendix A. The size of the calibration set considered here is 1000. We use separate or the same datasets for parameter tuning and calibration, respectively. For each experiment, the test set comprises the remaining data. By default, we use the APS score for experiments involving confidence calibration and conformal training.

**Tuning bias is not always negligible for the parameter tuning in conformal prediction.** In Figure 1, we present the tuning biases caused by various tuning methods using ResNet-18 on CIFAR-100. The results indicate that most methods do not introduce significant tuning biases, achieving similar coverage gaps regardless of whether the same dataset or separate datasets are used for calibration and tuning. For example, the coverage gaps of RAPS using the same and separate datasets are almost identical, as shown in Figure 1. However, in vector scaling and ConfTr (ft.), using the same datasets could result in much larger coverage gaps than using a hold-out dataset for tuning. We conjecture that these differences may be due to the large number of parameters tuned in these methods. In what follows, we provide in-depth analyses to explore the factors influencing tuning biases in these methods.

### 3.2. Parametric Scaling Laws of Tuning Bias

We conduct an additional analysis from two key perspectives: (1) the number of parameters and (2) the size of the calibration set. The experiments provide deeper insights into how tuning methods influence bias formation. By the convention to control the number of unfrozen parameters, we focus on confidence calibration using vector scaling. We could control the number of unfrozen parameters by freezing a portion of the parameters in vector scaling. We *freeze* a portion of the parameters in vector scaling. In detail, we do not let the proportion of the parameters be optimized randomly. The ratio of frozen parameters ranges from 0% to 40% by 10% increments. The detail of vector scaling is provided in Appendix A.2. We use separate or the same datasets for parameter tuning and calibration, respectively, with APS. All methods of conformal prediction are under the same error rate $\alpha = 0.1$. To ensure the reliability of our results, we repeat each experiment 30 times and report both the average result and its corresponding standard deviation.

**The tuning bias scales up with the complexity of parameter space.** We first consider the tuning bias under a different number of unfrozen parameters in Figure 2 (a).

As shown in Figure 2 (a), the tuning bias increases with the number of unfrozen parameters in vector scaling. With a larger number of parameters, the tuning bias increases by more than 50% compared to the lower number of parameter spaces. In general conclusions in learning theory, the higher number of parameters to tune means more complex parameter space without additional constraints. The more complex parameter space leads to more generalization bias. Here, our empirical results also support this conclusion: *the tuning bias scales with the complexity of parameter space*.

**The effect of calibration set size** Then, we consider the tuning bias under different sizes of the calibration set on Figure 2 (b). The calibration set size varies from 6000 to 10000 with an increment of 1000. The parameter tuning method used here is vector scaling with full parameter space. For the size of the calibration set, the tuning bias decreases with the size of the calibration set increasing, as shown in Figure 2 (b). The tuning bias decreases about 50% from the size of 6000 to 10000. As the size of the calibration set increases, the tuning bias approaches zero. The empirical results support the conclusion that *the tuning bias scales down with the size of the calibration set*.

### 3.3. Intuitive Explanation

Combining the above two empirical results, we could see the tuning bias scales up with the complexity of parameter space and down with the size of the calibration set empirically. Based on the framework of learnability of conformal prediction (Bai et al., 2022; Gupta et al., 2022), we could reinterpret the threshold calculation step Eq. (2) as solving a constrained empirical risk minimization (ERM) problem with a single learnable parameter:

$$\hat{t} = \arg\min_{t} \sum_{i \in \mathcal{I}_{\text{cal}}} |\widehat{\mathcal{C}}_t(\boldsymbol{x}_i)|/n$$

$$\text{such that } \sum_{i \in \mathcal{I}_{\text{cal}}} \mathbb{1}\{y_i \notin \widehat{\mathcal{C}}_t(\boldsymbol{x}_i)\}/n \leq \alpha$$

where $\mathcal{I}_{\text{cal}}$ is the index set of the hold-out set $\mathcal{D}_{\text{cal}}$. Then, the tuning parameter in the conformal prediction could be placed into the optimization problem with the learnable parameter $(\lambda, t)$. According to the general results in ERM theory, a more complex parameter space requires more data to learn the optimal tuning parameters. These results intuitively explain the phenomenon that appears above. Further, we will quantify the parametric scaling law of the tuning bias on the complexity of parameter space and the size of the calibration set, respectively, under the constrained ERM framework. To our knowledge, we are the first to explore this framework for the learnability of tuning parameters in the context of conformal prediction. Based on the learnability of tuning parameters in the context of conformal prediction, we further establish the theoretical results of the

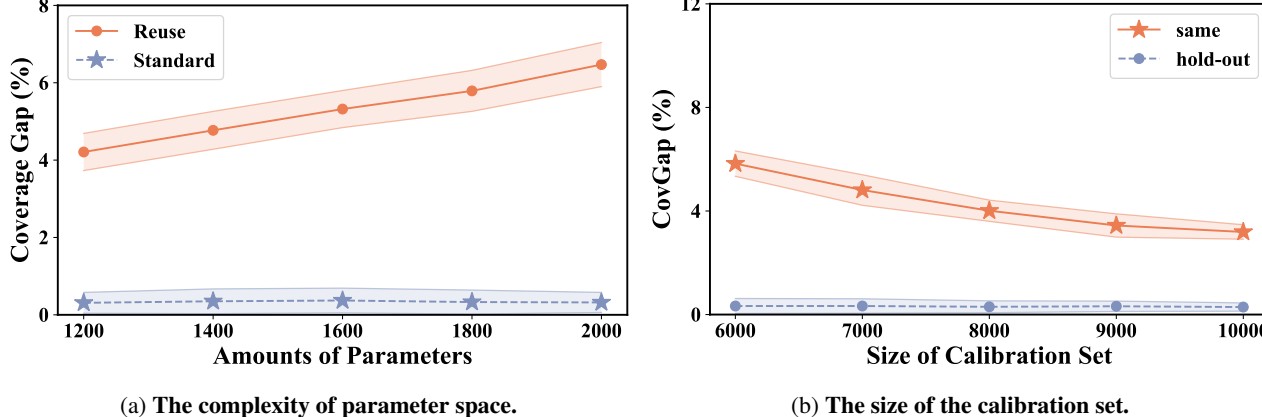

(a) **The complexity of parameter space.**      (b) **The size of the calibration set.**

*Figure 2.* **Parametric scaling law,** on (a) the complexity of parameter space and (b) the size of the calibration set, using ResNet-18 on ImageNet using APS. The calibration set size for Figure (a) is 6000. The 'hold-out' and 'same' denote that we use separate/same datasets for parameter tuning and calibration, respectively. *Tuning bias* is the difference between the coverage gaps of models tuned in the setting of *hold-out* and *same*. The tuning bias scales up with the number of parameters and scales down with the size of the calibration set.

tuning bias in the following section.

In summary, the empirical results reveal that the tuning bias is negligible for simple parameter tuning and scales up with the complexity of parameter space and down with the size of the calibration set. We intuitively explain the empirical results using the learnability of tuning parameters in the context of conformal prediction. Next, we will provide the theoretical results of the tuning bias in the following section.

# 4. Theoretical Results

In this section, we provide the theoretical results for the tuning bias. We first introduce the problem formulation in Section 4.1. We introduce the tuning bias in general cases. Then, we consider the theoretical results of the tuning bias in finite and infinite parameter cases in Section 4.2 and Section 4.3, respectively. The application of these results to some specific cases is also considered.

## 4.1. Problem Formulation

For a score function $S^\lambda$ with $\lambda \in \Lambda$, the prediction set is defined as:

$$\widehat{\mathcal{C}}_t^\lambda(\boldsymbol{x}) := \{y \in \mathcal{Y} : S^\lambda(\boldsymbol{x}, y) \le t\},$$

where $t$ is a threshold. Given a dataset $\mathcal{D}$, we denote the set of scores under parameter $\lambda$ as $\mathcal{S}_{\mathcal{D}}^\lambda := \{S^\lambda(\boldsymbol{x}_i, y_i) \mid (\boldsymbol{x}_i, y_i) \in \mathcal{D}\}$ and $\mathcal{S}_{\mathcal{D}} := \{S(\boldsymbol{x}_i, y_i) \mid (\boldsymbol{x}_i, y_i) \in \mathcal{D}\}$. The associated threshold for $\mathcal{S}_{\mathcal{D}}^\lambda$ is defined as:

$$\hat{t}_{\mathcal{D}}^\lambda = Q_{((1-\alpha)(1+1/n))}(\mathcal{S}_{\mathcal{D}}^\lambda), \qquad (4)$$

where $Q_p(\mathcal{S})$ denotes the $p$-th empirical quantile of non-empty set $\mathcal{S}$: $Q_p(\mathcal{S}) = \inf\{q : |\{s \in \mathcal{S} : s \le q\}| \ge p|\mathcal{S}|\}$.

We formulate the parameter tuning using the same set as the calibration set for conformal prediction as follows: the tuning parameter $\hat{\lambda}$ is selected by minimizing a pre-specified loss function $\ell$ on the calibration set $\mathcal{D}_{\text{cal}}$:

$$\hat{\lambda} := \hat{\lambda}_{\mathcal{D}_{\text{cal}}} = \arg\min_{\lambda \in \Lambda} \ell(\lambda, \mathcal{S}_{\mathcal{D}_{\text{cal}}}^\lambda).$$

The conformal prediction set with parameter tuning is reformulated as:

$$\mathcal{C}(\boldsymbol{x}) := \widehat{\mathcal{C}}_{\hat{t}_{\mathcal{D}_{\text{cal}}}^{\hat{\lambda}}}^{\hat{\lambda}}(\boldsymbol{x}). \qquad (5)$$

**Tuning bias**    While this procedure appears straightforward, using the same set for both parameter tuning and conformal prediction introduces a tuning bias. To understand this phenomenon, let us examine why the standard split conformal prediction works. The validity guarantee in Eq. (1) relies on the exchangeability assumption - specifically, the score function $S$ is independent of the calibration set $\mathcal{D}_{\text{cal}}$. This independence ensures that the score of a test sample $S(\boldsymbol{x}_{\text{test}}, y_{\text{test}})$ is exchangeable with the scores of calibration samples $\{S(\boldsymbol{x}_i, y_i) \mid (\boldsymbol{x}_i, y_i) \in \mathcal{D}_{\text{cal}}\}$. However, if we reuse the same set for both parameter tuning and calibration for conformal prediction, this crucial exchangeability property is violated. The transformed score $S^{\hat{\lambda}}$ depends on the calibration set $\mathcal{D}_{\text{cal}}$ through the parameter selection process. Consequently, the scores of test samples lose their exchangeability with the scores of calibration samples, invalidating the coverage guarantee in Eq. (1). To understand this phenomenon mathematically, we demonstrate the coverage gap for the conformal prediction with parameter tuning by decomposing the coverage gap into two components:

**Theorem 4.1** (Coverage gap for conformal prediction with parameter tuning). *If the samples in $\mathcal{D}_{cal} \cup \mathcal{D}_{test}$ are independent and identically distributed, then the coverage gap*

*of the conformal prediction set with parameter tuning $\widehat{\mathcal{C}}$ defined in Eq.* (5) *are bounded as follows:*

$$CovGap(\mathcal{C}) \leq \mathbb{E}\mathfrak{R}_\Lambda + \varepsilon_{\alpha,n},$$

*where $\varepsilon_{\alpha,n} := \lceil (1+n)(1-\alpha) \rceil / n - (1-\alpha) \geq 0$ is the coverage gap of the standard split conformal prediction set with the size of the calibration set $n$ and error rate $\alpha$, and the term $\mathfrak{R}_\Lambda$ with $\mathcal{T} \subset \mathbb{R}$,*

$$\mathfrak{R}_\Lambda := \sup_{\lambda \in \Lambda, t \in \mathcal{T}} \left| \frac{1}{n} \sum_{i \in [n]} \mathbb{1}\{S^\lambda(\boldsymbol{x}_i, y_i) \leq t\} \right.$$
$$\left. - \mathbb{P}\left( S^\lambda(\boldsymbol{x}_{test}, y_{test}) \leq t \mid \mathcal{D}_{cal} \right) \right|.$$

The proof of Theorem 4.1 is provided in Appendix E. The upper bound of the coverage gap could be decomposed into two components: $\mathbb{E}\mathfrak{R}_\Lambda$ and $\varepsilon_{\alpha,n}$. The first term, $\mathbb{E}\mathfrak{R}\Lambda$, introduces an additional bound for the coverage gap, which is not present in the standard conformal prediction framework, containing only the second term, $\varepsilon_{\alpha,n}$. As the definition of tuning bias in Definition 2.3, we could conclude that

$$TuningBias(\mathcal{C}) \leq \mathbb{E}\mathfrak{R}_\Lambda.$$

In fact $\mathbb{E}\mathfrak{R}_\Lambda$ is closely related to the classical empirical process theory (van der Vaart & Wellner, 1996; Vershynin, 2018). We could define the class associated with $\mathfrak{R}_\Lambda$ as:

$$\mathcal{H}_\Lambda = \left\{ \mathbb{1}\{S^\lambda(\boldsymbol{x}, y) \leq t\}, \lambda \in \Lambda, t \in \mathbb{R} \right\}.$$

The bound $\mathbb{E}\mathfrak{R}_\Lambda$ is the empirical process on the class $\mathcal{H}_\Lambda$:

$$\mathbb{E}\mathfrak{R}_\Lambda = \mathbb{E} \sup_{g \in \mathcal{H}_\Lambda} \left| \frac{1}{n} \sum_{i \in [n]} g(\boldsymbol{x}_i, y_i) - \mathbb{E}[g(\boldsymbol{x}_{test}, y_{test}) \mid \mathcal{D}_{cal}] \right|.$$

It depends on the complexity of parameter space $\Lambda$. Theorem 4.1 is a general result that applies to any parameter space $\Lambda$ without any constraint. Next, we provide the bound analysis of the tuning bias in finite and infinite parameter space and consider several applications using these results.

### 4.2. Tuning Bias in Finite Parameter Space

When the parameter space $\Lambda$ is finite, we bound the tuning bias using classical concentration inequality, Dvoretzky–Kiefer–Wolfowitz inequality (Dvoretzky et al., 1956), on the empirical process with the finite union of probability:

**Proposition 4.2.** *For a finite parameter space $\Lambda$, we have*

$$TuningBias(\mathcal{C}) \leq \sqrt{\frac{\log(2|\Lambda|)}{2n}} + \frac{1}{\sqrt{2n}\sqrt{\log(2|\Lambda|)}}.$$

The proof of Proposition 4.2 is provided in Appendix F. The result in Proposition 4.2 is a general result that applies to

any finite parameter space $\Lambda$. The examples include parameter tuning in some specific score functions (Angelopoulos et al., 2021; Huang et al., 2024), score selection (Yang & Kuchibhotla, 2024) and aggregation (Luo & Zhou, 2024) with finite candidates, and early stopping in conformal prediction (Liang et al., 2023) and so on.

**Application to parameter tuning in RAPS score** The Regularized Adaptive Prediction Sets (RAPS) method (Angelopoulos et al., 2021) introduces a score function that requires tuning two hyperparameters:

$$S_{\text{RAPS}}(\boldsymbol{x}, y, p) = S_{\text{APS}}(\boldsymbol{x}, y, p) + \gamma \cdot (o(y, \boldsymbol{x}, p) - k_{\text{reg}})^+,$$

where $S_{\text{APS}}$ is defined in Eq. (9), $o(y, \boldsymbol{x}, p)$ represents the rank of probability $p(y|\boldsymbol{x})$ among all possible classes, $p(y'|\boldsymbol{x})$ ($y' \in [K]$), and $\gamma$ and $k_{\text{reg}}$ are hyperparameters. The notation $(z)^+$ signifies the positive part of $z$. The RAPS method optimize the set *size* or *adaptiveness* with hyperparameters $\lambda = (\gamma, k_{\text{reg}})^\top$ from a predetermined finite parameter space with $|\Lambda_{\text{RAPS}}| = M \cdot \bar{K}$ where $M$ is the number of grid points for $\gamma$, and $\bar{K}$ is the pre-specified max number of classes (Angelopoulos et al., 2021). The tuning bias is bounded by Proposition 4.2 as follows:

**Corollary 4.3.** *For a classification problem and the holdout set size $n$, the tuning bias of RAPS is bounded by:*

$$TuningBias(\mathcal{C}_{RAPS}) \leq \sqrt{\frac{\log(2M\bar{K})}{2n}} + \frac{1}{\sqrt{2n}\sqrt{\log(2M\bar{K})}}.$$

The proof of Corollary 4.3 is provided in Appendix G. Although the RAPS method (Angelopoulos et al., 2021) recommends using an additional hold-out set for hyperparameter selection, our findings allow for direct hyperparameter selection without the need for an extra hold-out set. As for SAPS (Huang et al., 2024), the parameter space is also finite, and we could also apply the same analysis to SAPS.

**Application to score selection with finite candidate** Considering there are $M$ candidate score functions $\{S_m, m \in [M]\}$ for constructing conformal prediction sets, let $\mathcal{C}_m(\boldsymbol{x})$ be the prediction set obtained using score function $S_m$ at a given error rate $\alpha$. The optimal score function is selected by minimizing the empirical size of the prediction set:

$$\hat{m} = \underset{m \in \{1,\dots,M\}}{\arg\min} \sum_{i \in \mathcal{I}_{\text{cal}}} |\mathcal{C}_m(\boldsymbol{x}_i)| / |\mathcal{I}_{\text{cal}}|.$$

This selection process introduces a tuning bias that can be bounded using Proposition 4.2 with $|\Lambda| = M$. This case is also considered by the score selection issues (Yang & Kuchibhotla, 2024). They propose a score selection framework for constructing minimal conformal prediction regions by selecting optimal score functions from finite candidates.

Their work introduces two methods. The first optimizes prediction width using the same hold-out set as the calibration set of conformal prediction, and the second maintains coverage guarantees using a separate hold-out set. The first achieves minimal prediction interval width with approximate coverage guarantees. It provides a bound of coverage gap for the first method, as a special case of our result:

**Corollary 4.4.** *For a classification problem, and the hold-out set size $n$, the tuning bias of selection of score functions (Yang & Kuchibhotla, 2024) with the number of candidates $M$ is bounded by:*

$$TuningBias(\mathcal{C}) \leq \sqrt{\frac{\log(2M)}{2n}} + \frac{1}{\sqrt{2n}\sqrt{\log(2M)}}.$$

The proof of Corollary 4.4 is provided in Appendix G. For the score aggregation (Luo & Zhou, 2024), we analyze it in Appendix L. Next, we extend our analysis to an infinite parameter space and apply it to conformal prediction with confidence calibration, where the parameter space is infinite.

### 4.3. Tuning Bias in Infinite Parameter Space

Before introducing the infinite parameter case, we first introduce the concept of Vapnik-Chervonenkis (VC) dimension (Vapnik & Chervonenkis, 1971). VC-dimension is a measure of the complexity of classes of Boolean functions.

**Definition 4.5** (VC Dimension). Let $\mathcal{F}$ be a class of Boolean functions defined on a domain $\Omega$, $f : \Omega \to \{0, 1\}$ for all $f \in \mathcal{F}$. A subset $\Omega' \subseteq \Omega$ is said to be *shattered* by $\mathcal{F}$ if any function $g : \Omega' \to \{0, 1\}$ can be obtained by restricting some function $f \in \mathcal{F}$ onto $\Omega'$. The VC dimension of $\mathcal{F}$, denoted by $\text{VC}(\mathcal{F})$, is the size of the largest subset $\Omega' \subseteq \Omega$ that can be shattered by $\mathcal{F}$. If the size of the largest shattered subset is infinite, we say the VC dimension is infinite.

Although the concept may seem hard to understand, it can be viewed as the maximum number of points that the class of functions can shatter intuitively. With a more complex class of functions, it is more likely to shatter a larger number of points. It is a measure of the complexity of the class of functions, $\mathcal{F}$. The intuition is that the larger the complexity of the class of functions, the larger the bound of the empirical process is. By the definition of $\mathcal{H}$ and Lemma D.6 in Appendix D, we develop a general bound of the tuning bias:

**Proposition 4.6.** *For an infinite parameter space $\Lambda \subset \mathbb{R}^d$, we have the tuning bias is bounded by:*

$$TuningBias(\widehat{\mathcal{C}}) \leq C\sqrt{\frac{d+1}{n}}$$

*for some universal constant $C > 0$.*

The proof of Proposition 4.6 is provided in Appendix H. The bound in Proposition 4.6 is a general result that applies

to any infinite parameter space $\Lambda \subset \mathbb{R}^d$. It shows that the tuning bias increases with the dimension of parameter space $d$ and decreases with the size of the hold-out set $n$. Next, we will present an application for confidence calibration.

**Application to confidence calibration** Consider a classification problem with $K$ classes, where we have two confidence calibration methods: Temperature scaling (TS) and vector scaling (VS) (Dabah & Tirer, 2024; Guo et al., 2017; Xi et al., 2024). The details of temperature scaling and vector scaling with tuning parameter $\lambda$ are given in Appendix A.2. For the two methods, there is a significant difference in the dimension of parameter space $\Lambda$: For temperature scaling, the parameter space is $\Lambda_{\text{TS}} = \mathbb{R}^+$, containing only one positive real number $\lambda$. For vector scaling, the parameter space is $\Lambda_{\text{VS}} = \mathbb{R}^K \times \mathbb{R}^K$, containing $2K$ parameters. Since vector scaling has a more complex parameter space compared to temperature scaling, the tuning bias of vector scaling is larger than that of temperature scaling:

**Corollary 4.7.** *For a classification problem with $K$ classes, the calibration set size $n$ and the same pre-trained model, the tuning bias of temperature scaling is smaller than that of vector scaling: $TuningBias(\mathcal{C}_{TS}) \leq TuningBias(\mathcal{C}_{VS})$, where $\mathcal{C}_{TS}$ and $\mathcal{C}_{VS}$ are the prediction sets of temperature scaling and vector scaling, respectively.*

The proof of Corollary 4.7 is provided in Appendix I. The result shows that the tuning bias of temperature scaling is smaller than that of vector scaling. We provide a result for the binary classification case in Section 5, where the tuning bias of temperature scaling is zero.

In this section, we derive the upper bounds of tuning bias in finite and infinite spaces, proving the parametric scaling law of tuning bias, and provide examples including RAPS, score selection and confidence calibration. Next, we discuss how to reduce the tuning bias with two specific cases.

## 5. Discussion of Potential Solutions

In the previous analysis, we demonstrate the parametric scaling law of tuning bias in relation to parameter space complexity and calibration set size. Here, we provide an initial discussion of potential solutions to mitigate the tuning bias that arises from using the same data for both calibration and parameter tuning. It is important to note that the common practice is to use separate hold-out datasets for calibration and parameter tuning. Mitigating tuning bias can be particularly beneficial in real-world applications when splitting the dataset is impractical due to data scarcity.

Following the scaling law, we can implement two principles to reduce the tuning bias: increasing the size of calibration sets or reducing the complexity of tuning parameters. The former indicates that we can decrease the bias by col-

lecting more data points for the calibration set, but it is challenging in data-scarce scenarios. We then focus on the latter - reducing the complexity of tuning parameters, which is commonly implemented to address overfitting issues. In the following, we discuss this principle with two special cases of parameter tuning in split conformal prediction.

*Table 1.* **Tuning bias comparison of various methods.** The calibration set size is 2500 for CIFAR-100 and 5000 for ImageNet using APS. Results in **bold** indicate superior performance.

| Methods | CIFAR-100 (%) | ImageNet (%) |
|---|---|---|
| Temperature scaling | **0.14 ± 0.01** | **0.04 ± 0.01** |
| Vector scaling | 1.13 ± 0.02 | 6.63 ± 0.03 |
| ConfTr (ft.) w/ OP | **0.52 ± 0.37** | **0.40 ± 0.31** |
| ConfTr (ft.) w/o OP | 6.15 ± 0.86 | 21.68 ± 0.58 |

**Reducing the parameter number**    A straightforward way to constrain parameter complexity is to reduce the number of parameters in model tuning. Here, we provide an example of parameter reduction through weight sharing. In particular, we show the benefits of weight sharing by comparing the tuning biases of temperature scaling (TS) and vector scaling (VS), where the former can be viewed as a special case of vector scaling with a shared parameter $|\Lambda_{TS}| < |\Lambda_{VS}|$. The empirical results in Table 1 show that TS with fewer parameters achieves much smaller tuning biases than VS, which is also supported by Figure 1 and Figure 3. Theoretically, we provide a theoretical result for the comparison in the case of binary classification:

**Proposition 5.1.** *For a binary classification problem, we have TuningBias($\mathcal{C}_{TS}$) = 0 ≤ TuningBias($\mathcal{C}_{VS}$), where $\mathcal{C}_{TS}$ and $\mathcal{C}_{VS}$ are the prediction sets of models tuned by temperature scaling and vector scaling, respectively.*

We provide the proof of the above proposition in Appendix J. From the example, we demonstrate that designing tuning algorithms with few parameters will be beneficial in preventing the tuning bias. Also, we analyze the relationship between the number of parameters and tuning bias by freezing different numbers of parameters within VS. The results in Figure 2 (a) show that increasing the number of parameters leads to higher tuning bias, supporting the claim. However, reducing the parameter number may severely degrade the performance of parameter tuning, making it inapplicable in some scenarios, which motivates us to apply regularization to the tuning algorithm.

**Regularization in the tuning**    Another effective approach is the use of regularization methods, which introduce additional constraints or penalties to model parameters, predictions, or loss functions. Here, we present an example of applying an order-preserving constraint to model predic-

tions: $\forall (\boldsymbol{x}, y), o(y, \boldsymbol{x}, p) = o(y, \boldsymbol{x}, p_\lambda)$ where $o(y, \boldsymbol{x}, p)$ is the order of the true label $y$ in the sorted predicted probabilities $p(\cdot|\boldsymbol{x})$, and $p_\lambda$ is the probability after transformation $\lambda$. Taking ConfTr (Stutz et al., 2022) as the tuning algorithm on the final linear layer, we present the following proposition to show the benefit of the order-preserving constraint.

**Proposition 5.2.** *For a classification problem with $K$ classes, and the calibration set size $n$, we have the tuning bias of $\mathcal{C}_{op}$ is less than the one of $\mathcal{C}$: TuningBias($\mathcal{C}_{op}$) ≤ TuningBias($\mathcal{C}$), where $\mathcal{C}$ and $\mathcal{C}_{op}$ represent the prediction sets of models tuned by ConfTr, with $\mathcal{C}_{op}$ under the order-preserving constraint and $\mathcal{C}$ without it.*

The proof of Proposition 5.2 is provided in Appendix K. And by Lemma K.1, the dimension of parameter space of order-preserving vector scaling is 2, which is much smaller than that without constraint, $2K$. By Lemma K.3, the dimension of parameter space of ConfTr with order-preserving constraint is $K + 2$, which is much smaller than that of matrix scaling with its dimension being $K^2 + K$. We also verify the argument by the empirical results shown in Table 1. The tuning bias of ConfTr with order-preserving constraint is much smaller than that without it. This approach is advantageous as it can generalize to complex paradigms of parameter tuning (e.g., fine-tuning the final layer). It is worth noting that designing tailored regularization techniques for specific tuning methods in conformal prediction may be a promising direction for future research.

# 6. Related Works

**Conformal prediction**    Conformal prediction is a method for uncertainty quantification that ensures that the prediction intervals or sets cover the true label with a user-defined error rate (Papadopoulos, 2008; Vovk, 2012; Vovk et al., 2005). On the one hand, conditional coverage validity is a key property of conformal prediction, ensuring the algorithm fairness (Angelopoulos et al., 2021; Gibbs et al., 2024; Huang et al., 2024; Romano et al., 2020). On the other hand, to enhance the efficiency of conformal prediction, recent studies (Kiyani et al., 2024a;b; Liu et al., 2025; Stutz et al., 2022) have proposed training-based methods, which could be regarded as adapters, performing tuning prior to conformal prediction. Beyond these, with many non-conformity scores proposed, the score selection or aggregation is another challenge in conformal prediction (Fan et al., 2024; Gasparin & Ramdas, 2024; Ge et al., 2024; Luo & Zhou, 2024; Qin et al., 2024; Yang & Kuchibhotla, 2024). These recent paradigms of conformal prediction typically require tuning parameters on a hold-out dataset, and our work provides the first study to quantify the negative effect induced by using the same dataset for tuning and calibration.

**Learnability** As classical learnability theory, a learnable model could be regarded as a risk minimization model with a specific hypothesis class (van der Vaart & Wellner, 1996; Vapnik, 1991; 1999; Vapnik & Chervonenkis, 1971; Vershynin, 2018). And further, the constrained ERM is developed as a special case of learnability theory, where the hypothesis class is the set with a specific constraint or structure, such as structural risk minimization of character recognition (Guyon et al., 1991), of data-dependent hypothesis class (Shawe-Taylor et al., 1998), of Rademacher penalty method (Koltchinskii, 2001) and rough set-based classifier (Liu et al., 2020). As for conformal prediction, several works study learn the conformal prediction directly by designing a specific objective function (Noorani et al., 2024; Stutz et al., 2022). Further, the conformal prediction is a special case of risk minimization problem with the constraint of coverage guarantee, and the generalization loss for the size of prediction set could be quantified (Gupta et al., 2022; Yang & Kuchibhotla, 2024), including classification (Bai et al., 2022) and regression (Fan et al., 2024; Gupta et al., 2022). As we proposed in Section 4, we interpret the tuning bias as a further constrained risk minimization problem of conformal prediction and provide the first systematic theoretical analysis of the tuning bias, offering a novel tool for analyzing the learnability of general conformal prediction problems.

## 7. Conclusion

In this paper, we first uncover an unobservable phenomenon related to the tuning bias in conformal prediction with parameter tuning. Empirically, we observe that the tuning bias is negligible when the parameter space's complexity is low, and the hold-out set is sufficiently large. Our theoretical findings establish a parametric scaling law for the tuning bias. Additionally, we explore potential solutions to address the implications of our theoretical results, emphasizing strategies to mitigate tuning bias by reducing the complexity of the parameter space. For future research, incorporating the structure of the parameter space could improve the precision of our theoretical results.

**Limitation** Our theoretical results provide a unified framework for understanding tuning bias - a general phenomenon in conformal prediction. However, the bound of tuning bias is not tight without more assumptions on the modelling. The bound could be more precise with more assumptions on the modelling. Here, we provide an example of regularization on the learnability for conformal prediction, which roughly tightens tuning bias. The tuning bias could be more precise under more assumptions on the modelling, such as the structure of the score and transformation.

## Acknowledgement

We thank the anonymous reviewers for their valuable comments and suggestions. This research is supported by the Shenzhen Fundamental Research Program (Grant No.JCYJ202308007091809020), the SUSTech-NUS Joint Research Program and in part by the National Natural Science Foundation of China (NSFC) under grant 12371290. We gratefully acknowledge the support of the Center for Computational Science and Engineering at the Southern University of Science and Technology.

## Impact Statement

This paper presents work whose goal is to advance the field of Machine Learning. There are many potential societal consequences of our work, none which we feel must be specifically highlighted here.

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

# A. Basic Experimental Setup

In this section, we provide a detailed experimental setup. In Appendix A.1, we describe the pre-trained models. In Appendix A.2, we describe the parameter tuning processes for confidence calibration, score function tuning, score aggregation and ConfTr.

## A.1. Pre-trained Models

We conduct this empirical study on three image classification benchmarks: CIFAR-10, CIFAR-100, and ImageNet. For CIFAR-10, we use a simple CNN consisting of three convolutional layers and two fully connected layers. For CIFAR-100 and ImageNet, we employ ResNet-18 (He et al., 2016). The ImageNet experiments utilize pre-trained classifiers available in TorchVision (Paszke et al., 2019), while the CIFAR-10 and CIFAR-100 classifiers are trained from scratch using their respective full training sets. For CIFAR-100, the network is trained for 200 epochs using SGD with a momentum of 0.9, a weight decay of 0.0005, and a batch size of 256. The initial learning rate is set to 0.1 and decreases by a factor of 5 at epochs 60, 120, and 160. Similarly, for CIFAR-10, the network is trained for 120 epochs using SGD with identical momentum, weight decay, and batch size settings. The initial learning rate is also set to 0.1 and decreases by a factor of 5 at epochs 30, 60, and 90.

## A.2. Setup for Parameter Tuning

**Confidence calibration**  We investigate two commonly used confidence calibration methods, including temperature scaling and vector scaling (Dabah & Tirer, 2024; Guo et al., 2017; Xi et al., 2024). Temperature scaling is the simplest extension for Platt scaling (Platt, 1999) and is a calibration method that adjusts the model's output logits through a scalar parameter $\lambda$. The transformed output probability is defined as a special case of vector scaling with tuning parameter $\lambda \in \mathbb{R}^+$:

$$p_\lambda^{\text{TS}}(y \mid \boldsymbol{x}) = \psi_y(f(\boldsymbol{x})/\lambda), \tag{6}$$

where $f(\boldsymbol{x})$ is the original logit output from the pre-trained deep classifier model in Section A.1, and $\psi_y$ denotes the $y$-th element of the softmax function:

$$\psi_j(z) = \frac{\exp(z_j)}{\sum_{k=1}^K \exp(z_k)}, \quad \text{for } j = 1, \dots, K, \tag{7}$$

where $z \in \mathbb{R}^K$ is the input logit vector and $z_j$ is the $j$-th component of the input logit vector and $\psi_j(z)$ represents the $j$-th component of the output probability vector. The vector scaling (Guo et al., 2017) for a logits output of the pre-trained deep classifier model $f(\boldsymbol{x}) \in \mathbb{R}^K$ with classes $K$ is defined as:

$$p_\lambda^{\text{VS}}(y \mid \boldsymbol{x}) = \psi_y(W \circ f(\boldsymbol{x}) + b), \tag{8}$$

where $\lambda = (W^\top, b^\top)^\top \in \mathbb{R}^{2K}$, the notation "$\circ$" means element-wise product of elements in two vectors. We optimize the parameter $\lambda$ using the objective negative log-likelihood (NLL) to obtain the optimal parameter

$$\ell_{\text{NLL},p}(\lambda) = -\frac{1}{n} \sum_{(\boldsymbol{x},y) \in \mathcal{D}_{\text{cal}}} \log(p_\lambda(y \mid \boldsymbol{x})),$$

where $p_\lambda(y \mid \boldsymbol{x})$ is the transformed output probability from temperature scaling or vector scaling. The transformed score function $S^\lambda(X, y)$ is a function of $p(y|X)$ based on some base score function, such as APS (Romano et al., 2020) defined as:

$$S_{\text{APS}}(\boldsymbol{x}, y, p) = \sum_{y_i \in \mathcal{Y}} p(y_i|\boldsymbol{x}) \cdot \mathbb{I}\{p(y_i|\boldsymbol{x}) > p(y|\boldsymbol{x})\} + u \cdot p(y|\boldsymbol{x}) \tag{9}$$

given probability output $p := p(y|\boldsymbol{x})$ for a classification task, and $u$ is a random variable drawn from the uniform distribution $U[0, 1]$, and THR (Sadinle et al., 2019) defined as:

$$S_{\text{THR}}(\boldsymbol{x}, y, p) = 1 - p(y \mid \boldsymbol{x}). \tag{10}$$

**Score function tuning**  Score function tuning is a common step in the pipeline of conformal prediction. In this study, we focus on two of the most widely used score functions: RAPS (Angelopoulos et al., 2021) and SAPS (Huang et al., 2024):

$$S_{\text{SAPS}}(\boldsymbol{x}, y, p) := \begin{cases} u \cdot p_{\max}(\boldsymbol{x}), & \text{if } o(y, \boldsymbol{x}, p) = 1, \\ p_{\max}(\boldsymbol{x}) + (o(y, p(\boldsymbol{x})) - 2 + u) \cdot \gamma, & \text{else,} \end{cases}$$

where $p_{\max}(\boldsymbol{x})$ is the maximum predicted probability among all classes for input $\boldsymbol{x}$, $o(y, \boldsymbol{x}, p)$ is the order of the true label $y$ in the sorted predicted probabilities $p(y|\boldsymbol{x})$, and $\gamma$ is a tuning parameter.

A two-stage grid search for parameter searching is performed for both RAPS and SAPS. Initially, the search is conducted over the range $[0.01, 0.3]$ with a step size of $0.01$ to determine the optimal parameter $x$. Subsequently, a finer search is carried out within $[x - 0.005, x + 0.005]$ using a granularity of $0.001$. Notably, since RAPS is not sensitive to its parameter $k_{\text{reg}}$, we focus on $\gamma$.

**Score aggregation**  Score aggregation could significantly enhance the performance of conformal predictors. Recent studies (Fan et al., 2024; Gasparin & Ramdas, 2024; Ge et al., 2024; Luo & Zhou, 2024; Qin et al., 2024; Yang & Kuchibhotla, 2024) have proposed aggregation techniques that combine multiple non-conformity scores with tunable weights, which can enhance the efficiency of conformal predictors. These methods typically require a hold-out set to search for the optimal weights through grid search. In this study, we investigate the aggregation of three score functions—APS, RAPS, and SAPS—all derived from the same pre-trained classifier as a case example. Specifically, we use three non-negative parameters $w_1$, $w_2$, and $w_3$, which represent the tunable weights for aggregating the three score functions. These parameters satisfy the constraint $w_1 + w_2 + w_3 = 1$, and we evaluate all possible value combinations with a granularity of $0.1$. The parameters $w_1$, $w_2$, and $w_3$ are restricted to the set $\{0, 0.1, 0.2, \ldots, 1\}$. We enumerate all triplets $(w_1, w_2, w_3)$ such that:

$$w_1 + w_2 + w_3 = 1, \quad w_1, w_2, w_3 \geq 0, \quad \text{and} \quad w_1, w_2, w_3 \in \{0, 0.1, 0.2, \ldots, 1\}.$$

This approach ensures that all feasible combinations of $w_1$, $w_2$, and $w_3$ are comprehensively explored under the given constraint and granularity.

**C-Adapter**  Conformal Adapter (C-Adapter) (Liu et al., 2025) is an adapter-based tuning method designed to enhance the efficiency of conformal predictors without sacrificing accuracy. It achieves this by appending an adapter layer to pre-trained classifiers for conformal prediction. With C-Adapter, the model tends to produce extremely high non-conformity scores for incorrect labels, thereby improving the efficiency of prediction sets across various coverage rates.

For tuning the C-Adapter, we use the Adam optimizer (Kingma & Ba, 2017) with a batch size of 256 and a learning rate of 0.1. The model is tuned for 10 epochs, and the only parameter, $T$, is set to $1 \times 10^{-4}$ by default. In our empirical study, we explore the application of C-Adapter using hold-out data.

**Conformal Training**  Conformal Training (ConfTr) (Stutz et al., 2022) is a training framework designed to enhance the efficiency of conformal predictors. Its loss function is defined as:

$$\mathcal{L}_{\text{ConfTr}}(f(\boldsymbol{x}; \boldsymbol{\theta}), y, \tau_\alpha^{\text{soft}}) = \mathcal{L}_{\text{cls}}(f(\boldsymbol{x}; \boldsymbol{\theta}), y) + \lambda \mathcal{L}_{\text{size}}(f(\boldsymbol{x}; \boldsymbol{\theta}), \tau_\alpha^{\text{soft}}),$$

where $\mathcal{L}_{\text{cls}}$ represents the classification loss, $\mathcal{L}_{\text{size}}$ approximates the size of the prediction set at a coverage rate of $1 - \alpha$, $\tau_\alpha^{\text{soft}}$ denotes the soft threshold, and $\lambda$ controls the strength of the regularization term. Here, $\tau_\alpha^{\text{soft}}$ denotes the soft threshold and $\lambda$ controls the strength of the regularization term. To improve efficiency, ConfTr adjusts the training objective by incorporating a regularization term that minimizes the average prediction set size at a specified error rate.

Beyond training models from scratch, ConfTr can also function as a post-training adjustment method, allowing fine-tuning of the fully connected layer or an appended layer to enhance further the efficiency of conformal predictors (Huang et al., 2023; Stutz et al., 2022). We denote this version of ConfTr as **ConfTr (ft.)**. In our empirical study, we explore this application of ConfTr (ft.) using hold-out data. Specifically, we append a fully connected layer to the trained classifier and fine-tune only the parameters of this layer using the Adam optimizer (Kingma & Ba, 2017) with a learning rate of 0.001. The hyperparameter $\alpha$ is set to 0.01 by default. For $\lambda$, we use a value of 0.0001.

# B. Additional Experimental Results

We investigate various conformal prediction algorithms that require parameter tuning. This section provides additional results for these methods. Typically, we vary the size of the calibration set to examine the CovGap. For vector scaling, we consider both the size of the calibration set and the number of parameters (i.e., the amount of frozen weights).

By the results in this section, it is worthy noting that the parameter numbers of those tuning methods (e.g., VS and C-Adapter) are positively related to the class numbers of the dataset. Thus, datasets with more classes require more parameter numbers

in the tuning, leading to a larger tuning bias. This explains why those methods perform differently in various datasets. The details are provided as following.

### B.1. Additional Results for Confidence Calibration

**Temperature scaling**   Additional experimental results on the tuning bias with temperature scaling are shown in Figures 3a and 3b, using APS and THR as the score functions, respectively.

**Vector scaling**   Figure 3c and Figure 3d present additional experimental results on the tuning bias with vector scaling, using THR as the score function. The experiments vary the size of the calibration set and the number of parameters.

The results in Figure 3 show that the tuning bias of temperature scaling is small for all datasets and score functions under various settings. The tuning bias of vector scaling is generally large except for CIFAR-10. For comparison, the tuning bias of temperature scaling is generally smaller than that of vector scaling.

### B.2. Additional Results for Score Function Tuning

We focus on two widely used score functions: RAPS and SAPS. The results for them are provided in Figures 4a and 4b. The results in these figures shows the tunig bias of RAPS and SAPS is not significantly different from the one with a hold-out set under various settings.

### B.3. Additional Results for Score Aggregation

In this study, we investigate the aggregation of three score functions—APS, RAPS, and SAPS—all derived from the same pre-trained classifier as a case example. The results are provided in Figure 4c. The results in this figure shows the tuning bias of score aggregation is not significantly different from the one with a hold-out set under various settings.

### B.4. Additional Results for C-Adapter

Additional experimental results on CovGap with C-Adapter are shown in Figures 4d and 4e, using APS and THR as the score functions, respectively. The results in these figures shows the tuning bias of C-Adapter is not significantly different from the one with a hold-out set under various settings.

### B.5. Additional Results for ConfTr (ft.)

Additional experimental results on CovGap with ConfTr (ft.) are shown in Figures 5a and 5b, using APS and THR as the score functions, respectively. The results in these figures shows the tuning bias of ConfTr (ft.) is significantly different from the one with a hold-out set under various settings except CIFAR-10.

## C. Additional Results for Regression

We clarify that our analysis reveals a general phenomenon of conformal prediction in both classification and regression tasks. Section 3 focuses on classification, as many parameter tuning methods (like TS/VS and ConfTr) are designed for classification tasks. Here, we provide the results for regressions to show the case of regression tasks following previous work (Liang et al., 2024) with a target coverage of 90% and 30 repetitions. The details are presented as following:

We conduct an empirical study to evaluate methods for constructing prediction intervals using the Conformalized Quantile Regression (CQR) framework. The experiments are performed on the Protein Structure dataset obtained from the UCI repository, which comprises $N = 45730$ data points with feature dimension $d = 9$. The base learners employed are `RandomForestQuantileRegressor` instances from Python's `quantile_forest` package. The target nominal coverage level for all experiments is set to $1 - \alpha = 0.9$.

We investigate two distinct strategies for utilizing the data reserved after model training (referred to as calibration set and hold-out set): **Same**: A single dataset is allocated for *both* model selection (i.e., choosing the optimal hyperparameter configuration from a set of candidates) and the subsequent calibration step (i.e., determining the empirical quantile required to adjust the prediction interval width); **Hold-out**: The reserved data is partitioned into two disjoint sets of equal size, where the first set is used for model selection, while the second set is used for calibration, consequently requiring twice the amount

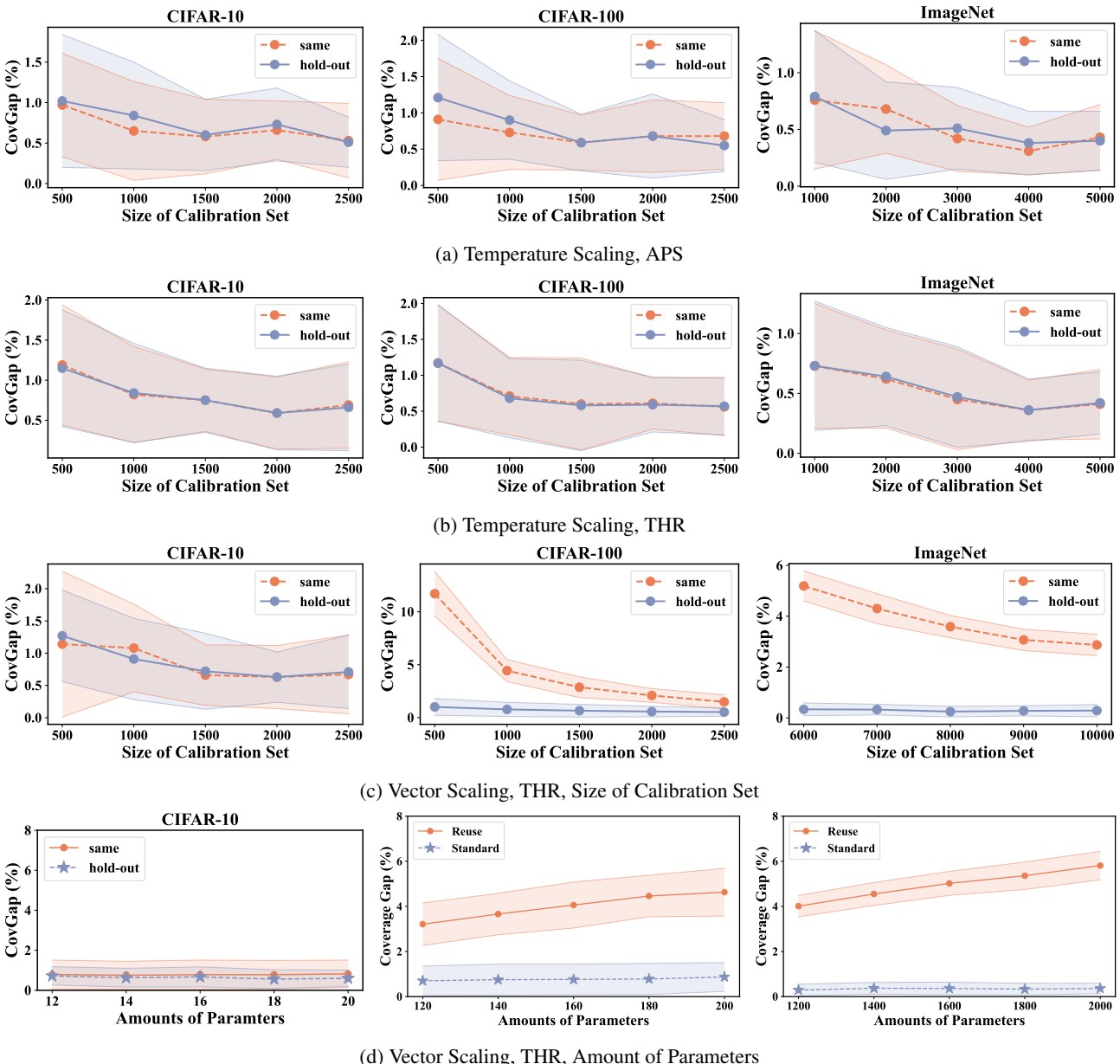

(a) Temperature Scaling, APS

(b) Temperature Scaling, THR

(c) Vector Scaling, THR, Size of Calibration Set

(d) Vector Scaling, THR, Amount of Parameters

*Figure 3.* **Additional results for confidence calibration across various datasets and score functions.** The coverage gaps are obtained on the conformal prediction with a ***hold-out*** set or the ***same*** set as calibration set to parameter tuning. Figures 3a and 3b show the tuning bias with temperature scaling, using APS and THR as the score functions, respectively. The coverage gaps are obtained on the conformal prediction with a ***hold-out*** set or the ***same*** set as calibration set to parameter tuning. Figures 3c and 3d show the tuning bias with vector scaling, using THR as the score function. The experiments vary the size of the calibration set and the number of parameters. The experiments are conducted on pre-trained models in Appendix A.1. The coverage rate is set to 0.9. The figure shows that the tuning bias of temperature scaling is small for all datasets and score functions under various settings. The tuning bias of vector scaling is generally large except CIFAR-10. For comparison, the tuning bias of temperature scaling is generally smaller than that of vector scaling.

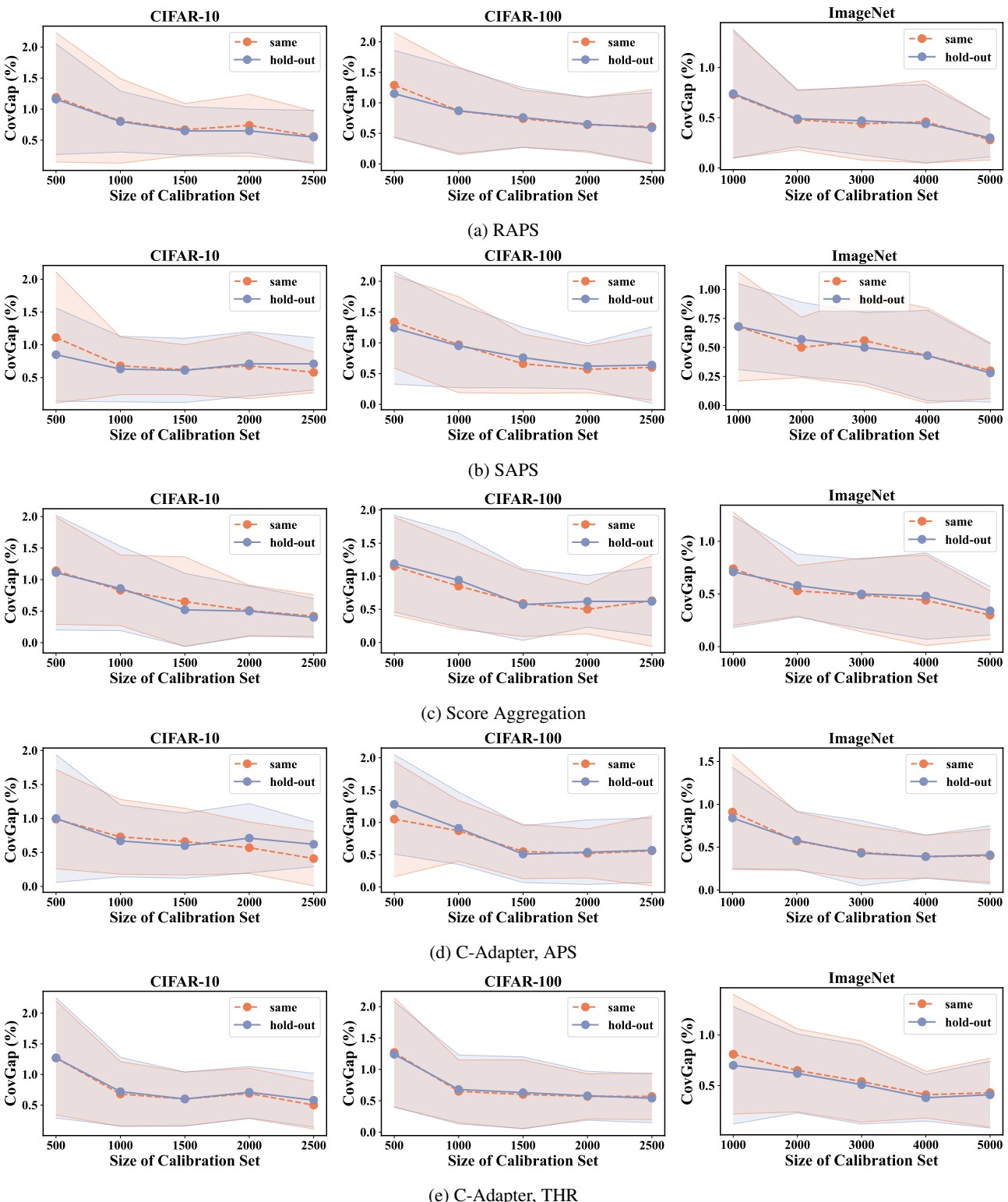

(a) RAPS

(b) SAPS

(c) Score Aggregation

(d) C-Adapter, APS

(e) C-Adapter, THR

*Figure 4.* **Additional results for score function tuning, aggregation, and C-Adapter across various datasets and score functions.** The coverage gaps are obtained on the conformal prediction with a ***hold-out*** set or the ***same*** set as calibration set to parameter tuning. Figures 4a and 4b show the tuning bias with RAPS and SAPS, respectively. Figure 4c shows the tuning bias with score aggregation. Figures 4d and 4e show the tuning bias with C-Adapter, using APS and THR as the score functions, respectively. The experiments are conducted on pre-trained models in Appendix A.1. The coverage rate is set to 0.9. The figure shows that the tuning bias of RAPS, SAPS, score aggregation, and C-Adapter is not significantly different from the one with a hold-out set under various settings.

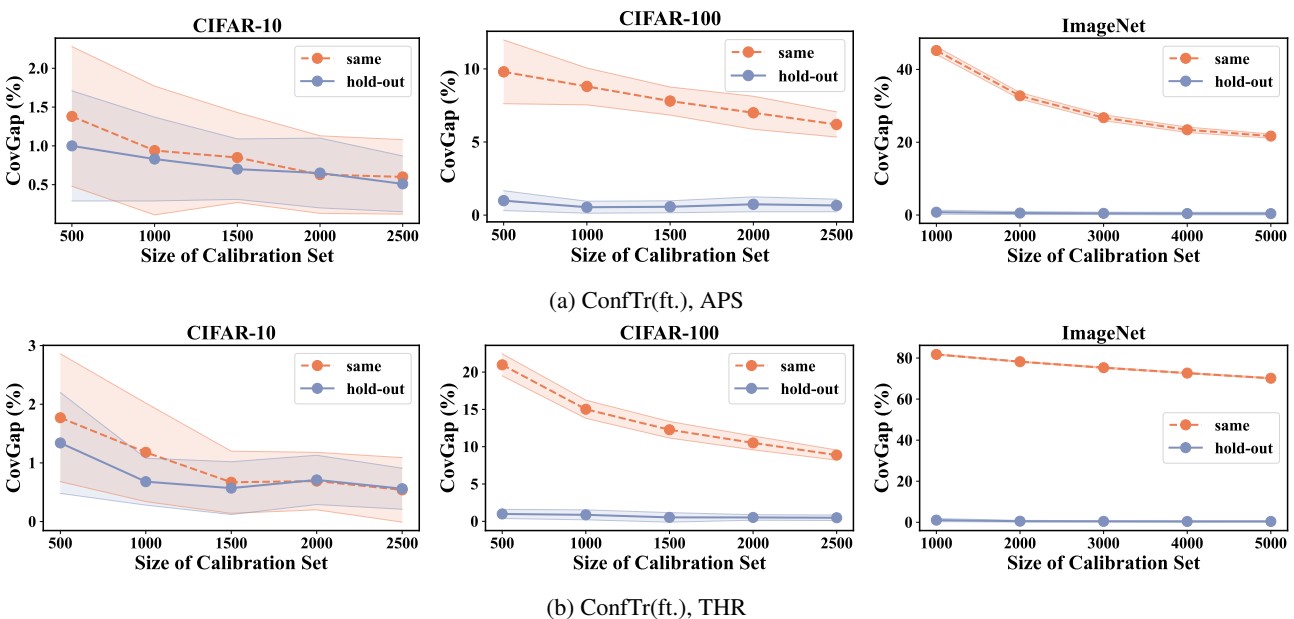

(a) ConfTr(ft.), APS

(b) ConfTr(ft.), THR

*Figure 5.* **Additional results for ConfTr(ft.) across various datasets and score functions.** The coverage gaps are obtained on the conformal prediction with a ***hold-out*** set or the ***same*** set as calibration set to parameter tuning. Figures 5a and 5b show the tuning bias with ConfTr(ft.), using APS and THR as the score functions, respectively. The experiments are conducted on pre-trained models in Appendix A.1. The coverage rate is set to 0.9. The figure shows that the tuning bias of ConfTr (ft.) is generally small for all datasets and score functions under various settings.

of reserved data compared to the Shared strategy to achieve an equivalently sized dataset for the final calibration phase. Here we consider the following two experiments:

**Experiment 1: Varying the number of candidate models.** For the first experiment, the size of calibration set in the Shared strategy is fixed at $n = 50$. For the Shared strategy, these 50 data points are employed for both model selection and calibration, while for the Split strategy, 50 data points are used for model selection, and a separate set of 50 data points is used for calibration, amounting to 100 reserved data points in total. The complexity of the model selection task is modeled by varying the number of candidate models. All candidate models are `RandomForestQuantileRegressor` instances. The `max_features` parameter is varied across 10 distinct values, specifically `np.linspace(d/10, d, 10)`. The `n_estimator` parameter is adjusted to alter the total pool of models: We vary the number of candidate models by adjusting the `n_estimator` parameter to 4, 8, 12, 16, and 20 values, which combined with the 10 distinct `max_features` values results in 40, 80, 120, 160, and 200 candidate models respectively.

**Experiment 2: Varying the Calibration Set Size** In this experiment, the number of candidate models is held constant at 200 (derived from 20 `n_estimator` values and 10 `max_features` values). We vary the size of the calibration dataset, denoted as $n$. For the **Same** strategy, $n$ represents the size of the single dataset used for both model selection and calibration, while for the **Hold-out** strategy, $n$ denotes the size of *each* of the two datasets (one for selection and one for calibration), resulting in a total reserved data of $2 \times n$ for the **Hold-out** strategy. The values investigated for both strategies are $n \in \{100, 200, 300, 400, 500\}$.

The results of the two experiments are summarized in Table 2 and Table 3, respectively. Mean Coverage and Std Coverage are calculated from the 30 independent runs. The results of the two experiments are similar to the ones in the main text.

## D. Some Useful Lemmas and Corollaries

In this section, we provide some useful lemmas and corollaries that are used in the main text.

**Lemma D.1.** *(Dvoretzky et al., 1956; Massart, 1990, Dvoretzky–Kiefer–Wolfowitz Inequality) Let $\boldsymbol{x}_1, \boldsymbol{x}_2, \dots, \boldsymbol{x}_n$ be independent and identically distributed (i.i.d.) random variables with common distribution function $F$. The empirical*

*Table 2.* Results for varying the number of candidate models. For the Same strategy, calibration size $n = 50$. For the Hold-out strategy, $n = 50$ for model selection and $n = 50$ for calibration. Target coverage $1 - \alpha = 0.9$.

| # Models | Method | Coverage Mean (Std) | Length Mean (Std) | Coverage Gap Mean (Std) | Tuning Bias Mean |
|---|---|---|---|---|---|
| 40 | Same | 0.8437 (0.0383) | 12.89 (1.22) | 0.0563 (0.0348) | 0.0246 |
| | Hold-out | 0.9251 (0.0268) | 17.28 (2.96) | 0.0317 (0.0212) | - |
| 80 | Same | 0.8229 (0.0354) | 12.41 (1.34) | 0.0771 (0.0353) | 0.0412 |
| | Hold-out | 0.9049 (0.0404) | 16.01 (2.51) | 0.0359 (0.0252) | - |
| 120 | Same | 0.8258 (0.0564) | 12.27 (1.63) | 0.0742 (0.0520) | 0.0324 |
| | Hold-out | 0.8953 (0.0480) | 15.94 (3.17) | 0.0418 (0.0293) | - |
| 160 | Same | 0.8373 (0.0467) | 12.40 (1.33) | 0.0627 (0.0413) | 0.0371 |
| | Hold-out | 0.9163 (0.0286) | 16.24 (2.06) | 0.0256 (0.0217) | - |
| 200 | Same | 0.8447 (0.0494) | 12.44 (1.41) | 0.0553 (0.0429) | 0.0187 |
| | Hold-out | 0.9037 (0.0406) | 15.09 (2.25) | 0.0366 (0.0246) | - |

*Table 3.* Results for varying the calibration set size ($n$). The number of candidate models is fixed at 200. For the Hold-out strategy, $n$ refers to the size of each of the two sets (selection and calibration), totaling $2n$ reserved points. Target coverage $1 - \alpha = 0.9$.

| $n$ | Method | Coverage Mean (Std) | Length Mean (Std) | Coverage Gap Mean (Std) | Tuning Bias Mean |
|---|---|---|---|---|---|
| 100 | Same | 0.8459 (0.0509) | 12.77 (1.25) | 0.0541 (0.0433) | 0.0150 |
| | Hold-out | 0.9079 (0.0456) | 16.16 (3.11) | 0.0391 (0.0278) | - |
| 200 | Same | 0.8553 (0.0378) | 13.10 (0.96) | 0.0448 (0.0337) | 0.0118 |
| | Hold-out | 0.9043 (0.0394) | 15.12 (1.72) | 0.0330 (0.0212) | - |
| 300 | Same | 0.8608 (0.0326) | 13.04 (0.93) | 0.0392 (0.0296) | 0.0158 |
| | Hold-out | 0.8967 (0.0323) | 14.46 (1.04) | 0.0234 (0.0192) | - |
| 400 | Same | 0.8595 (0.0292) | 13.13 (0.56) | 0.0405 (0.0266) | 0.0157 |
| | Hold-out | 0.8927 (0.0322) | 14.37 (1.16) | 0.0248 (0.0201) | - |
| 500 | Same | 0.8767 (0.0388) | 13.25 (0.68) | 0.0327 (0.0269) | 0.0037 |
| | Hold-out | 0.9007 (0.0323) | 14.21 (0.65) | 0.0290 (0.0189) | - |

*distribution function $F_n$ is defined by*

$$F_n(\boldsymbol{x}) = \frac{1}{n} \sum_{i=1}^{n} \mathbf{1}\{\boldsymbol{x}_i \leq \boldsymbol{x}\}.$$

*Then for any $\varepsilon > 0$, the following bound holds:*

$$\mathbb{P}\left(\sup_{\boldsymbol{x} \in \mathbb{R}^d} |F_n(\boldsymbol{x}) - F(\boldsymbol{x})| > \varepsilon\right) \leq 2\exp(-2n\varepsilon^2).$$

**Lemma D.2** (Gaussian Tail Inequality). *For any $a > 0$, we have:*

$$\int_a^\infty e^{-t^2}\, dt \leq \frac{e^{-a^2}}{2a}$$

*Proof.* For any $a > 0$, we have:

$$
\begin{aligned}
\int_a^\infty e^{-t^2}\, dt &= \int_a^\infty \frac{t}{t} e^{-t^2}\, dt \\
&= \frac{1}{2} \int_a^\infty \frac{1}{t} \cdot 2t e^{-t^2}\, dt \\
&\leq \frac{1}{2a} \int_a^\infty 2t e^{-t^2}\, dt \\
&= \frac{e^{-a^2}}{2a}
\end{aligned}
$$

The inequality in the third line is because when $t \geq a > 0$, $\frac{1}{t} \leq \frac{1}{a}$. $\qquad\square$

**Lemma D.3.** *For a score function $S^\lambda$ with one-dimensional parameter space $\Lambda = \mathbb{R}$, consider the class:*

$$\mathcal{H}_\Lambda = \{\mathbf{1}\{S^\lambda(\boldsymbol{x}, y) \leq t\} \mid \lambda \in \mathbb{R}, t \in \mathbb{R}\}.$$

*If $S^\lambda(\boldsymbol{x}, y)$ is continuous, bounded over $\lambda$ for any fixed $(\boldsymbol{x}, y)$, then $VC(\mathcal{H}_\Lambda) \leq 2$. Specifically, if $S^\lambda(\boldsymbol{x}, y)$ is distinct over $\mathcal{X} \times \mathcal{Y}$ for any fixed $\lambda$, then $VC(\mathcal{H}_\Lambda) = 2$.*

*Proof.* We prove that $VC(\mathcal{H}_\Lambda) \leq 2$ in two steps:

1. We show that $VC(\mathcal{H}_\Lambda) \geq 2$. Given any two distinct points $(\boldsymbol{x}_1, y_1)$ and $(\boldsymbol{x}_2, y_2)$, we need to verify that all four possible binary labellings (0,0), (0,1), (1,0), (1,1) can be realized by some choice of $\lambda$ and $t$. Consider $S^\lambda(\boldsymbol{x}, y)$:

   - For labeling (0,0): Choose $t$ small enough such that both points have values greater than $t$.
   - For labelling (1,1): Choose $t$ large enough such that both points have values less than or equal to $t$.
   - Then forlabellingss (1,0) or (0,1): Since the points are distinct and $S^\lambda(\boldsymbol{x}, y)$ is continuous in $\lambda$ over $\mathbb{R}$, by the intermediate value theorem, there exists some $\lambda$ where $S^\lambda(\boldsymbol{x}_1, y_1) \neq S^\lambda(\boldsymbol{x}_2, y_2)$. Then, choose $t$ between these two values.

   Therefore, $VC(\mathcal{H}_\Lambda) \geq 2$.

2. We prove that $VC(\mathcal{H}_\Lambda) \leq 2$. Suppose for contradiction that three distinct points can be shattered. Consider the labelling pattern (1,0,1). This would require some $\lambda$ and $t$ such that:

$$S^\lambda(\boldsymbol{x}_1, y_1) \leq t, \quad S^\lambda(\boldsymbol{x}_2, y_2) > t, \quad S^\lambda(\boldsymbol{x}_3, y_3) \leq t$$

implying $S^\lambda(\boldsymbol{x}_2, y_2) > \max\{S^\lambda(\boldsymbol{x}_1, y_1), S^\lambda(\boldsymbol{x}_3, y_3)\}$. However, since $\lambda$ is one-dimensional and $S^\lambda(\boldsymbol{x}, y)$ is continuous in $\lambda$, such a complex ordering cannot be achieved at a single $\lambda$ value for all possible three-point configurations. This phenomenon is because the continuity of $S^\lambda(\boldsymbol{x}, y)$ over $\mathbb{R}$ ensures that any crossing of function values must occur in pairs, making it impossible to maintain the required ordering consistently. Therefore, three points cannot be shattered, and $VC(\mathcal{H}_\Lambda) \leq 2$.

Combining both results, we conclude that $VC(\mathcal{H}_\Lambda) = 2$. $\square$

**Corollary D.4.** *For a score function $S^\lambda$ with parameter space $\Lambda = \mathbb{R}^+$, consider the class:*

$$\mathcal{H}_\Lambda = \{\mathbb{1}\{S^\lambda(\boldsymbol{x}, y) \leq t\} \mid \lambda \in \mathbb{R}^+, t \in \mathbb{R}\}.$$

*If $S^\lambda(\boldsymbol{x}, y)$ is continuous and bounded over $\lambda$ for any fixed $(\boldsymbol{x}, y)$ and $S^\lambda(\boldsymbol{x}, y)$ is distinct over $\mathcal{X} \times \mathcal{Y}$ for any fixed $\lambda$, then $VC(\mathcal{H}_\Lambda) = 2$.*

*Proof.* The proof mirrors Lemma D.3, with $\lambda > 0$. This restriction does not alter the VC dimension: First, we show that $VC(\mathcal{H}_\Lambda) \geq 2$. Two points can still achieve all labellings with positive $\lambda$ and $t$. Next, we show that $VC(\mathcal{H}_\Lambda) \leq 2$. The restriction to $\lambda > 0$ does not allow shattering three points. Thus, the VC dimension is 2. $\square$

**Lemma D.5.** *For a score function $S^\lambda$ with $d$-dimensional parameter space $\Lambda = \mathbb{R}^d$, consider the class:*

$$\mathcal{H}_\Lambda = \{\mathbb{1}\{S^\lambda(x, y) \leq t\} \mid \lambda \in \mathbb{R}^d, t \in \mathbb{R}\}.$$

*If $S^\lambda(x, y)$ is continuous and bounded over $\lambda$ for any fixed $(x, y)$, then $VC(\mathcal{H}_\Lambda) \leq d + 1$.*

*Proof.* We prove that $VC(\mathcal{H}_\Lambda) \leq d + 1$. Suppose for contradiction that $d + 2$ distinct points can be shattered. Consider any labelling pattern that requires one point to have a value strictly larger than $d + 1$ other points at the same $\lambda$. This would require some $\lambda$ and $t$ such that:

$$S^\lambda(x_i, y_i) \leq t \text{ for } i \in I, \quad S^\lambda(x_j, y_j) > t \text{ for some } j \notin I$$

where $|I| = d + 1$. This implies $S^\lambda(\boldsymbol{x}_j, y_j) > \max_{i \in I}\{S^\lambda(\boldsymbol{x}_i, y_i)\}$. However, since $\lambda$ is $d$-dimensional and $S^\lambda(\boldsymbol{x}, y)$ is continuous in $\lambda$, such a complex ordering requiring one point to dominate $d + 1$ other points cannot be achieved at a single $\lambda$ value. This phenomenon is because the continuity of $S^\lambda(\boldsymbol{x}, y)$ over $\mathbb{R}^d$ ensures that the ordering relationships between $d + 2$ points cannot be arbitrary with only $d$ degrees of freedom. Therefore, $d + 2$ points cannot be shattered, and $VC(\mathcal{H}_\Lambda) \leq d + 1$. $\square$

**Lemma D.6** (Empirical Process Bound via VC Dimension)**.** *For the class $\mathcal{H}_\Lambda$ with finite VC dimension $VC(\mathcal{H}_\Lambda) \geq 1$, we have*

$$\mathbb{E}\mathfrak{R}_{\Lambda, \mathcal{D}_{cal}} \leq C\sqrt{\frac{VC(\mathcal{H}_\Lambda)}{n}},$$

*where $C$ is a universal constant.*

*Proof.* It is a direct result of the VC dimension bound in Vershynin (2018, Theorem 8.3.23). $\square$

**Lemma D.7.** *If $S^\lambda(\boldsymbol{x}, y)$ is order-preserving for any transformation $\lambda \in \Lambda$, i.e., $S^\lambda(\boldsymbol{x}, y) \leq S^\lambda(\boldsymbol{x}', y')$ if and only if $S(\boldsymbol{x}, y) \leq S(\boldsymbol{x}', y')$ for any transformation $\lambda$, then for $(X, y)$ with the same distribution as $(X_i, y_i) \in \mathcal{D}_{cal}$*

$$\mathbb{P}(S^\lambda(\boldsymbol{x}, y) \leq \hat{t}^\lambda_{\mathcal{D}_{cal}} \mid \mathcal{D}_{cal}) = \mathbb{P}(S(\boldsymbol{x}, y) \leq \hat{t} \mid \mathcal{D}_{cal})$$

*for $\lambda \in \Lambda$ where $\hat{t}$ is defined in Eq. (2).*

*Proof.* Since $S^\lambda(\boldsymbol{x}, y)$ is order-preserving for any transformation $\lambda \in \Lambda$, we have

$$S^\lambda(\boldsymbol{x}, y) \leq S^\lambda(\boldsymbol{x}', y') \quad \text{if and only if} \quad S(\boldsymbol{x}, y) \leq S(\boldsymbol{x}', y')$$

for any transformation $\lambda$. By the definition of $\hat{t}(\lambda, \mathcal{D}_{cal})$ and $\hat{t}$, the position of $S^\lambda(\boldsymbol{x}, y)$ among $S^\lambda(\boldsymbol{x}_i, y_i)$ is the same as the position of $S(\boldsymbol{x}, y)$ among $S(\boldsymbol{x}_i, y_i)$. Then, we have

$$\mathbb{P}(S^\lambda(\boldsymbol{x}, y) \leq \hat{t}(\lambda, \mathcal{D}_{cal}) \mid \mathcal{D}_{cal}) = \mathbb{P}(S(\boldsymbol{x}, y) \leq \hat{t} \mid \mathcal{D}_{cal}).$$

$\square$

**Lemma D.8.** *For temperature scaling in binary classification, we have $p^{TS}_\lambda(y|X)$ is order-preserving over $(\boldsymbol{x}, y)$ for any transformation $\lambda \in \Lambda$.*

*Proof.* Noting that

$$p_\lambda^{\text{TS}}(y \mid \boldsymbol{x}) = \psi_y \left( \frac{f(\boldsymbol{x})}{\lambda} \right)$$

where $\psi_y$ is the softmax function defined in Eq. (7). Since

$$\psi_y(X) = \frac{\exp\left(f_y(\boldsymbol{x})\right)}{\exp\left(f_y(\boldsymbol{x})\right) + \exp\left(f_{1-y}(\boldsymbol{x})\right)} = \frac{1}{1 + \exp\left(f_{1-y}(\boldsymbol{x}) - f_y(\boldsymbol{x})\right)}$$

for binary label $y \in \{0, 1\}$, we have

$$\psi_y(f(\boldsymbol{x})) \leq \psi_{y'}(f(\boldsymbol{x}'))$$

$$\iff \frac{1}{1 + \exp\left(f_{1-y}(\boldsymbol{x}) - f_y(\boldsymbol{x})\right)} \leq \frac{1}{1 + \exp\left(f_{1-y'}(\boldsymbol{x}') - f_{y'}(\boldsymbol{x}')\right)}$$

$$\iff \exp\left(f_{1-y}(\boldsymbol{x}) - f_y(\boldsymbol{x})\right) \geq \exp\left(f_{1-y'}(\boldsymbol{x}') - f_{y'}(\boldsymbol{x}')\right)$$

$$\iff \exp\left((f_{1-y}(\boldsymbol{x}) - f_y(\boldsymbol{x}))/\lambda\right) \geq \exp\left((f_{1-y'}(\boldsymbol{x}') - f_{y'}(\boldsymbol{x}'))/\lambda\right), \forall \lambda > 0$$

$$\iff \psi_y\left(\frac{f(\boldsymbol{x})}{\lambda}\right) \leq \psi_{y'}\left(\frac{f(\boldsymbol{x}')}{\lambda}\right)$$

for any transformation $\lambda > 0$. $\qquad\square$

# E. Proof of Theorem 4.1

*Proof.* For any sample $(\boldsymbol{x}, y)$ with the same distribution as $(\boldsymbol{x}_i, y_i) \in \mathcal{D}_{\text{cal}}$ and independent of $\mathcal{D}_{\text{cal}}$, recalling the definition of $\mathfrak{R}_{\Lambda, \mathcal{D}_{\text{cal}}}$ in Theorem 4.1:

$$\mathfrak{R}_{\Lambda, \mathcal{D}_{\text{cal}}} := \sup_{\lambda \in \Lambda, t \in \mathcal{T}} \left| \frac{1}{n} \sum_{i \in [n]} \mathbb{1}\{S^\lambda(\boldsymbol{x}_i, y_i) \leq t\} - \mathbb{P}\left(S^\lambda(\boldsymbol{x}_{\text{test}}, y_{\text{test}}) \leq t \mid \mathcal{D}_{\text{cal}}\right) \right|$$

we have

$$\left| \frac{1}{n} \sum_{i=1}^n \mathbb{1}\{S^\lambda(\boldsymbol{x}_i, y_i) \leq t\} - \mathbb{P}(S^\lambda(\boldsymbol{x}, y) \leq t \mid \mathcal{D}_{\text{cal}}) \right| \leq \mathfrak{R}_{\Lambda, \mathcal{D}}$$

Then, we have

$$\frac{1}{n} \sum_{i=1}^n \mathbb{1}\{S^\lambda(\boldsymbol{x}_i, y_i) \leq t\} - \mathfrak{R}_{\Lambda, \mathcal{D}} \leq \mathbb{P}(S^\lambda(\boldsymbol{x}, y) \leq t \mid \mathcal{D}_{\text{cal}}) \leq \frac{1}{n} \sum_{i=1}^n \mathbb{1}\{S^\lambda(\boldsymbol{x}_i, y_i) \leq t\} + \mathfrak{R}_{\Lambda, \mathcal{D}}.$$

Letting $t = \hat{t}_{\mathcal{D}_{\text{cal}}}^\lambda$ defined in Eq. (4) and noting the truth that

$$\frac{1}{n} \sum_{i=1}^n \mathbb{1}\{S^\lambda(\boldsymbol{x}_i, y_i) \leq \hat{t}(\lambda, \mathcal{D}_{\text{cal}})\} = \frac{\lceil (1-\alpha)(1+n) \rceil}{n},$$

we have

$$\frac{\lceil (1-\alpha)(1+n) \rceil}{n} - \mathfrak{R}_{\Lambda, \mathcal{D}} \leq \mathbb{P}(S^\lambda(\boldsymbol{x}, y) \leq \hat{t}_{\mathcal{D}_{\text{cal}}}^\lambda \mid \mathcal{D}_{\text{cal}}) \leq \frac{\lceil (1-\alpha)(1+n) \rceil}{n} + \mathfrak{R}_{\Lambda, \mathcal{D}}.$$

Take expectations on both sides; we have

$$\text{CovGap}(\mathcal{C}, \alpha) \leq \mathbb{E}\mathfrak{R}_{\Lambda, \mathcal{D}} + \varepsilon_{\alpha, n}.$$

$\qquad\square$

## F. Proof of Proposition 4.2

*Proof.* By Lemma D.1, we have

$$\mathbb{P}(\sup_{t \in \mathcal{T}} |\frac{1}{n} \sum_{i=1}^{n} \mathbb{1}\{S^\lambda(\boldsymbol{x}_i, y_i) \leq t\} - \mathbb{P}(S^\lambda(\boldsymbol{x}, y) \leq t \mid \mathcal{D}_{\text{cal}})| > u) \leq 2\exp(-2nu^2).$$

By the property of probability of finite union over $\lambda \in \Lambda$, we have

$$\mathbb{P}(\sup_{\lambda \in \Lambda, t \in \mathcal{T}} |\frac{1}{n} \sum_{i=1}^{n} \mathbb{1}\{S^\lambda(\boldsymbol{x}_i, y_i) \leq t\} - \mathbb{P}(S^\lambda(\boldsymbol{x}, y) \leq t \mid \mathcal{D}_{\text{cal}})| > u) \leq 2|\Lambda|\exp(-2nu^2),$$

i.e.,

$$\mathbb{P}(\mathfrak{R}_{\Lambda, \mathcal{D}_{\text{cal}}} > u) \leq 2|\Lambda|\exp(-2nu^2).$$

Noting that the bound is too large around 0, we take a departure from the bound into two parts from the point $\sqrt{\frac{\log(2|\Lambda|)}{2n}}$ where the probability is just 1. Before this point, the bound is too large, we do not use it. Then, the expectation of $\mathfrak{R}_{\Lambda, \mathcal{D}_{\text{cal}}}$ is bounded by two parts:

$$\begin{aligned}
\mathbb{E}[\mathfrak{R}_{\Lambda, \mathcal{D}_{\text{cal}}}] &= \int_0^\infty \mathbb{P}(\mathfrak{R}_{\Lambda, \mathcal{D}_{\text{cal}}} > u)\, du \\
&= \int_0^{\sqrt{\frac{\log(2|\Lambda|)}{2n}}} \mathbb{P}(\mathfrak{R}_{\Lambda, \mathcal{D}_{\text{cal}}} > u)\, du + \int_{\sqrt{\frac{\log(2|\Lambda|)}{2n}}}^\infty \mathbb{P}(\mathfrak{R}_{\Lambda, \mathcal{D}_{\text{cal}}} > u)\, du \\
&\leq \sqrt{\frac{\log(2|\Lambda|)}{2n}} + \int_{\sqrt{\frac{\log(2|\Lambda|)}{2n}}}^\infty 2|\Lambda|\exp(-2nu^2)\, du \\
&= \sqrt{\frac{\log(2|\Lambda|)}{2n}} + \frac{2|\Lambda|}{\sqrt{2n}} \int_{\sqrt{\log(2|\Lambda|)}}^\infty \exp(-t^2)\, dt \\
&\leq \sqrt{\frac{\log(2|\Lambda|)}{2n}} + \frac{2|\Lambda|}{\sqrt{2n}} \cdot \frac{1}{2|\Lambda|\sqrt{\log(2|\Lambda|)}} \\
&= \sqrt{\frac{\log(2|\Lambda|)}{2n}} + \frac{1}{\sqrt{2n}\sqrt{\log(2|\Lambda|)}}
\end{aligned}$$

where the second inequality is due to Lemma D.2. □

## G. Proof of Corollary 4.3 and 4.4

*Proof.* It is a direct application of Proposition 4.2. □

## H. Proof of Proposition 4.6

*Proof.* It is a direct application of Lemma D.5 and D.6. □

## I. Proof of Corollary 4.7

*Proof.* It is a direct application of Proposition 4.6. □

## J. Proof of Proposition 5.1

*Proof.* We consider a binary classification problem with temperature scaling and THR score applied. The probability $p_\lambda^{\text{TS}}(y \mid \boldsymbol{x})$ is order-preserving of the label for a given input $X$ for any $\lambda \in \mathbb{R}^+$, i.e, $\forall \lambda \in \mathbb{R}^+$,

$$p_\lambda^{\text{TS}}(y \mid \boldsymbol{x}) \leq p_\lambda^{\text{TS}}(y' \mid \boldsymbol{x}) \iff p(y \mid \boldsymbol{x}) \leq p(y' \mid \boldsymbol{x}). \tag{11}$$

Then, for any two samples $(\boldsymbol{x}, y)$ and $(\boldsymbol{x}', y')$, with the property of softmax function $\psi_y$ on binary classification (see Lemma D.8 in Appendix D), we have

$$p_\lambda^{\mathrm{TS}}(y \mid \boldsymbol{x}) \leq p_\lambda^{\mathrm{TS}}(y' \mid \boldsymbol{x}) \iff p(y \mid \boldsymbol{x}) \leq p(y' \mid \boldsymbol{x}). \tag{12}$$

Then, the direct result is that the order of score function is preserved, i.e.,

$$S_{\mathrm{THR}}(\boldsymbol{x}, y, p_\lambda^{\mathrm{TS}}) \leq S_{\mathrm{THR}}(\boldsymbol{x}', y', p_\lambda^{\mathrm{TS}})$$
$$\iff S_{\mathrm{THR}}(\boldsymbol{x}, y, p) \leq S_{\mathrm{THR}}(\boldsymbol{x}', y', p).$$

Therefore, the order of the scores on the dataset $\mathcal{D}_{\mathrm{cal}}$ is preserved for any transformation $\lambda \in \mathbb{R}^+$, i.e, the order of $\mathcal{S}_{\mathcal{D}_{\mathrm{cal}}}^\lambda$ is the same as $\mathcal{S}_{\mathcal{D}_{\mathrm{cal}}}$ for any $\lambda \in \mathbb{R}^+$. Then by Lemma D.7 in Appendix D, we have

$$\mathbb{P}\left(S^\lambda(\boldsymbol{x}, y) \leq \hat{t}_{\mathcal{D}_{\mathrm{cal}}}^\lambda \mid \mathcal{D}_{\mathrm{cal}}\right) = \mathbb{P}\left(S(\boldsymbol{x}, y) \leq \hat{t} \mid \mathcal{D}_{\mathrm{cal}}\right) \tag{13}$$

for any $\lambda \in \mathbb{R}^+$, which means $\mathrm{TuningBias}(\widehat{\mathcal{C}}_{\mathrm{TS}}) = 0 \leq \mathrm{TuningBias}(\widehat{\mathcal{C}}_{\mathrm{VS}})$.

$\square$

# K. Order-preserving Regularization and Proof of Proposition 5.2

**Order-preserving regularization**  As discussed in Section 5, applying the regularization on the conformal prediction with confidence calibration or fine-tuning methods could reduce the tuning bias significantly. Taking order-preserving regularization as an example, we modify the class associated with $\mathfrak{R}_\Lambda$ as regularization form:

$$\mathcal{H}_{\lambda,\mathrm{op}} = \{f_{\lambda,t} \in \mathcal{H}_\Lambda \text{ s.t. } o(y, \boldsymbol{x}, p) = o(y, \boldsymbol{x}, p_\lambda)\}, \tag{14}$$

where $o(y, \boldsymbol{x}, p)$ is the order of the true label $y$ in the sorted predicted probabilities $p(\cdot|\boldsymbol{x})$, and $p_\lambda$ is the probability after confidence calibration like temperature scaling or vector scaling. The class $\mathcal{H}_{\Lambda,\mathrm{op}}$ is a constraint version of $\mathcal{H}_\Lambda$ with a special regularization. With the constrained ERM with artificial order-preserving regularization, we could obtain a less tuning bias. The tuning bias of the conformal prediction with order-preserving regularization is bounded by the tuning bias of the conformal prediction without order-preserving regularization as Proposition 5.2.

*Proof of Proposition 5.2.* The ConfTr(op) is a regularization case of ConfTr, where the class $\mathcal{H}_{\Lambda,\mathrm{op}}$ is used as the constraint. Since the class $\mathcal{H}_{\Lambda,\mathrm{op}}$ defined in Eq. (14) is a subset of $\mathcal{H}_\Lambda$, we have

$$\mathbb{E} \sup_{f \in \mathcal{H}_{\Lambda,\mathrm{op}}} \left|\frac{1}{n} \sum_{i \in [n]} f(\boldsymbol{x}_i, y_i) - \mathbb{E}[f(\boldsymbol{x}_{\mathrm{test}}, y_{\mathrm{test}}) \mid \mathcal{D}_{\mathrm{cal}}]\right| \leq \mathbb{E} \sup_{f \in \mathcal{H}_\Lambda} \left|\frac{1}{n} \sum_{i \in [n]} f(\boldsymbol{x}_i, y_i) - \mathbb{E}[f(\boldsymbol{x}_{\mathrm{test}}, y_{\mathrm{test}}) \mid \mathcal{D}_{\mathrm{cal}}]\right|,$$

which means that

$$\mathrm{TuningBias}(\mathcal{C}_{\mathrm{ConfTr\,(op)}}) \leq \mathrm{TuningBias}(\mathcal{C}_{\mathrm{ConfTr}}).$$

$\square$

**Order-preserving regularization for vector scaling**  For the vector scaling defined in Eq. (8), because the softmax function is order-preserving, the order-preserving regularization on the output probability in Eq. (7) is equivalent to the order-preserving regularization on the input logits $f(\boldsymbol{x})$. If we apply the order-preserving regularization on the vector scaling, we should have for any logits vector $f(\boldsymbol{x})$,

$$\forall j, j' \in [K], f_j(\boldsymbol{x}) \leq f_{j'}(\boldsymbol{x}) \iff W_j f_j(\boldsymbol{x}) + b_j \leq W_{j'} f_{j'}(\boldsymbol{x}) + b_{j'},$$

where $f_j(\boldsymbol{x})$ is the $j$-th element of the logits vector $f(\boldsymbol{x})$, $W_j$ is the $j$-th element of $W$, and $b_j$ is the $j$-th element of $b$.

**Lemma K.1.** *Vector scaling is order-preserving if and only if $W_j = W_{j'} > 0$ for all $j, j' \in [K]$ and $b_j = b_{j'}$ for all $j, j' \in [K]$.*

*Proof.* ($\Leftarrow$) It is a direct result of the definition of vector scaling.

($\Rightarrow$) If vector scaling is order-preserving, then

$$\forall j, j' \in [K], f_j(\boldsymbol{x}) \le f_{j'}(\boldsymbol{x}) \iff W_j f_j(\boldsymbol{x}) + b_j \le W_{j'} f_{j'}(\boldsymbol{x}) + b_{j'}.$$

We first exclude the case when $W_j < 0$ for any $j \in [K]$: For any $W_j < 0$, and fixed $b_j, b_{j'}$, we could find small enough $f_j(\boldsymbol{x})$, such that $f_j(\boldsymbol{x}) < f_{j'}(\boldsymbol{x})$ and $W_j f_j(\boldsymbol{x}) + b_j > W_{j'} f_{j'}(\boldsymbol{x}) + b_{j'}$, which contradicts the order-preserving property. Therefore, we only consider the case when $W_j, W_{j'} > 0$ for all $j, j' \in [K]$. Without loss of generality, we assume $f_j(\boldsymbol{x}) \le f_{j'}(\boldsymbol{x})$ and $W_j < W_{j'}$, and let $f_{j'}(\boldsymbol{x}) = f_j(\boldsymbol{x}) + \epsilon$ with $\epsilon \ge 0$, then we have

$$W_j f_j(\boldsymbol{x}) + b_j \le W_{j'} f_{j'}(\boldsymbol{x}) + b_{j'} \iff (W_j - W_{j'}) f_j(\boldsymbol{x}) \le b_{j'} - b_j + W_{j'}\epsilon$$

Let $\varepsilon \to 0$, we have

$$(W_j - W_{j'}) f_j(\boldsymbol{x}) \le b_{j'} - b_j$$

hold for any $f_j(\boldsymbol{x})$. We could find a small enough $f_j(\boldsymbol{x})$, such that

$$(W_j - W_{j'}) f_j(\boldsymbol{x}) > b_{j'} - b_j$$

This contradicts the order-preserving property. Therefore, we must have $W_j = W_{j'} > 0$ for all $j, j' \in [K]$ and $b_j = b_{j'}$ for all $j, j' \in [K]$. $\square$

*Remark* K.2. By Proposition 5.2, if we apply the order-preserving regularization on the vector scaling, the parameter space of the vector scaling is reduced to $\Lambda = \mathbb{R}^2$.

**Order-preserving regularization for matrix scaling**   Here we consider a more complex case that the parameter space of the scaling is a matrix: $W f(\boldsymbol{x}) + b$, where $f$ be a logits value function for a classifier of $K$ classes, and $W$ is a matrix of size $K \times K$ and $b$ is a vector of size $K$. Here the dimension of the parameter space is $K^2 + K$. We apply the order-preserving regularization on the matrix scaling:

**Lemma K.3.** *Let $f$ be a logits value function for a classification of $K$ classes. The matrix scaling $g(x) = W f(x) + b$ is order-preserving if and only if $W$ has the form $W = aI + \mathbf{1}v^T$ for some scalar $a > 0$ and vector $v \in \mathbb{R}^K$, and $b$ is a constant vector (i.e., $b_j = b_{j'}$ for all $j, j' \in [K]$). Here, $I$ is the $K \times K$ identity matrix and $\mathbf{1}$ is the $K$-dimensional vector of all ones.*

*Remark* K.4. Here, we regard C-Adapter as a special case of matrix scaling with order-preserving regularization. The above proposition shows that the order-preserving regularization reduces the dimension of parameter space from $K^2 + K$ to $K + 2$, which is much smaller than VS with its dimension being $2K$. Based on the parametric scaling law (Section 4), we explain why C-Adapter can achieve lower tuning bias than VS. And Lemma K.1 is a special case of Lemma K.3.

## L. Additional Theoretical Results of Score aggregation

For a more general case about the selection of scores, we could also apply the same analysis to the score aggregation (Luo & Zhou, 2024). In the setting of score aggregation, they choose an aggregation weights vector $\boldsymbol{w}$ to aggregate multiple score functions:

$$S_{\boldsymbol{w}}(\boldsymbol{x}, y) = \sum_{m=1}^{M} w_m S_m(\boldsymbol{x}, y),$$

where $\boldsymbol{w} = (w_1, \ldots, w_M)^\top$ is the aggregation weights vector. Define the parameter space $\Lambda$ as the set of all possible aggregation weights vectors, i.e., $\Lambda = \mathcal{W} = \{\boldsymbol{w} \in \mathbb{R}^M : \sum_{m=1}^{M} w_m = 1, w_m \ge 0\}$. Then, for grid research, we could also apply Proposition 4.2 to bound its tuning bias to obtain a more similar result as Corollary 4.4.

