# OpenReview forum: "Parametric Scaling Law of Tuning Bias in Conformal Prediction"
_ICML.cc/2025/Conference — ICML 2025 poster_

### Official Review · Reviewer_jaH5 · 2025-03-10

**Overall Recommendation:** 4

**Summary:**

The manuscript explores the phenomenon of tuning bias in the field of conformal prediction, which is a statistical method used to ensure that prediction intervals or sets cover the true value with a specified probability. The focus is on how the tuning of parameters, when done on the same dataset used for calibration, affects the coverage accuracy of the prediction models. Key points from the paper include:

1.	Tuning Bias Definition and Impact: Tuning bias is defined as the coverage gap that arises when the same data set is used for both parameter tuning and calibration. The paper empirically demonstrates that this bias is generally negligible for simple parameter tuning across various conformal prediction methods.

2.	Parametric Scaling Law: The study observes that the magnitude of tuning bias increases with the complexity of the parameter space and decreases with the size of the calibration set. This relationship is formalized through the derivation of upper bounds on the tuning bias, which align with empirical observations.

3.	Theoretical Framework: A theoretical framework is established to quantify the tuning bias, using empirical process theory within the extended parameter space. This framework provides a rigorous basis for understanding and predicting the behavior of tuning bias under different conditions.

4.	Empirical Studies: The paper includes extensive empirical evaluations involving methods like RAPS, SAPS, score aggregation, and confidence calibration methods, using datasets like CIFAR-100 and applying models like ResNet-18. These studies confirm the scaling laws and the minimal impact of tuning bias under typical conditions.

5.	Reduction Strategies: Potential strategies to mitigate tuning bias are discussed, focusing on increasing calibration set size or reducing parameter space complexity. Practical challenges such as data scarcity are acknowledged, suggesting order-preserving regularization as a promising approach to manage tuning bias effectively.

6.	Contributions and Future Directions: The primary contributions are identifying the negligible effect of tuning bias in many scenarios, formalizing the parametric scaling law of tuning bias, and proposing theoretical models to understand and predict tuning bias. The paper suggests further research could explore structured parameter spaces to refine the precision of tuning bias predictions.

Overall, this paper provides a significant theoretical and empirical foundation for understanding tuning bias in conformal prediction, offering insights that can help in designing more reliable machine learning models, particularly in settings where rigorous uncertainty quantification is critical.

**Claims And Evidence:**

The exact definition of exchangeability appears very late in the paper, which can be formalized, highlighted a bit of earlier. Also, a clearer discussion on the relationship between exchangeability and the using of the same dataset for tuning and calibration can be further highlighted.

I am not sure if I am correct, but for many of previous works, it seems that scaling law typically indicates larger parametric space / data scales leads to improve performance or robustness. However, in this paper, the authors seems to understand the problem of conformal prediction from a generalization perspective, exploring the impacts of parameter space and data scales, which is quite a style of classical machine learning. So, I am concerned if the “Scaling Law” can be somewhat misleading in the title.

I am quite confusing about two seemingly conflicting claims: “the tuning bias is negligible for simple parameter tuning in conformal prediction methods” and “the parametric scaling law of the tuning bias increases with parameter space complexity and decreases with calibration set size”. From my understanding, the former states that the violation of exchangeability is not actually a big deal yet latter states that it is still have side impacts, especially influenced by the data scale and parameter scale. Hoping the authors could made some clarification to address my misunderstandings.

**Essential References Not Discussed:**

The authors fully review the existing literature with many concrete examples, therefore I think the reference is good enough.

**Experimental Designs Or Analyses:**

From Figure 1, it seems that the CovGap values are small except for VS and ConfTr ft. Therefore, why do not we just choose other CP methods or are their some particular interesting in using those methods with more hyper-parameters like VS and ConfTr?

**Methods And Evaluation Criteria:**

I think the authors could further clarify how the CovGap is computed in practice, which seems critical for their empirical analyses.  Also, for the potential solutions section, the mentioned methods are quite easy. However, for a theoretical style paper, maybe it is not a big deal.

**Other Comments Or Suggestions:**

NA

**Other Strengths And Weaknesses:**

It seems that the authors consider the situation where the same dataset is used for tuning and calibration, aiming to understanding its impacts on tuning bias. However, I am concerned that if it is truly difficult to have another hold-out set, as the validation dataset can be easily separated into two parts, one for tuning and one for calibration. I think the authors could further emphasize why the considered setup is pragmatic or this problem has some other interesting factors from an academic point of view.

**Questions For Authors:**

Please see the comments above.

**Relation To Broader Scientific Literature:**

This paper presents work whose goal is to advance the field of Machine Learning. There are many potential societal consequences of our work, none which I feel must be specifically highlighted.

**Theoretical Claims:**

I think the authors would like to take their works as a theoretical style paper, which is evident from its large amount of discussions on theoretical analysis and small amount of experimental verification.

I did not check every details within the theoretical analysis, but it seems like an applications of PAC framework for CP problem, where the dataset size is echoed by n and the parameter space is echoed by VC dimension. I may raise two questions, 1) what’s the challenges, uniqueness, and contributions of this work from a theoretical perspective, as I think this paper should be categorized as a theoretical analysis work. 2) Besides echoing the observations within Section 3, what other observations can we draw from the new theoretical analysis.

---

> ### Author Rebuttal · Authors · 2025-03-31
>
> Thanks for your positive review and insightful feedback.
>
> **1. Exchangeability definition/discussion**
>
> Thank you for the suggestion. We agree that introducing the exact definition earlier can improve the clarity. In the current version, we defaultly introduce Exchangeability as a well-known assumption of CP (with references). In the final version, we will add the exact definition of Exchangeability in Sec. 2, as the reviewer suggested. We present in Subsec. 4.1 (Lines 261-274) why using the same dataset violates the Exchangeability. We will improve the clarity of this part in the final version.
>
> **2. Clarification of the term "scaling law"**
>
> Thank you for raising the concern. Recently, the term "scaling law" frequently appears in the context of large language models - the performance scales up with parameter/data numbers. We want to clarify that the above scaling law is formally termed "neural scaling law" (see Wikipedia), a special case of empirical scaling law. Notably, the term 'scaling law' in deep learning refers to a broader concept that describes the relationships between functional properties of interest (such as tuning bias in our work) and characteristics of the model architecture or dataset (e.g., model size) [1]. To improve the clarity, we will add a concise description of "scaling law" with references in the final version.
>
> **3. Seemingly conflicting claims**
>
> Thank you for highlighting the potential misunderstanding. We want to clarify that 'simple parameter tuning' in the first claim refers to methods with few parameters, such as RAPS and TS. As Line 135, we excluded methods with a larger number of parameters, such as VS and the fine-tuned version of ConfTr. We use the first claim to demonstrate that tuning bias can be negligible in certain cases, which motivates the subsequent analysis. This does *not* conflict with the second claim regarding the parametric scaling law, which illustrates when tuning bias can be either small or large. To enhance clarity, we will revise the first claim to state: 'The tuning bias is not always significant for parameter tuning in ...'
>
> **4. Concerns on CovGap**
>
> CovGap is computed empirically as the absolute difference between the target coverage ($1-\alpha$) and the empirical coverage:
> $$
> |(1-\alpha) - \frac{1}{n'} \sum_{i=1}^{n'} \mathbb{1}(y_i \in \hat{C}(x_i))|,
> $$
> where $n'$ is the size of the test set, and $ \hat{C}(x) $ is the CP set for an input $x$. We will add a concise description of CovGap in the final version. As for the method, we present potential solutions for mitigating tuning bias as an extension (See response #1 to reviewer mAZq). We hope the guideline can inspire more future work to design specific methods for addressing this challenge.
>
> **5. Theoretical contribution and uniqueness**
>
> Thank you for the recognition. In this work, we formulate the tuning bias into ERM framework, and propose a general theory to bound the tuning bias by the PAC and empirical process theories, explaining the empirical scaling law. Then, we derive the bias bounds in finite and infinite parameter cases, respectively. In particular, we also provide the specific bounds of tuning bias in various tuning methods. Lastly, we provide the theoretical results to support the two practical guidelines addressing the challenge. We list the theoretical contribution as follows:
>
> 1. **Problem formulation**: This work is the first to formulate the "tuning bias" arising from dataset reuse, which provides a new direction for understanding non-exchangeable CP.
> 2. **CP-specific complexity analysis**: We derive the bounds of tuning bias with the complexity measures (e.g., VC dimension) of the *CP-specific hypothesis class*, which can be developed as a theoretical toolkit for refined bounds of general biases in CP.
> 3. **Analytical framework for tuning methods**: we establish a framework to derive the bias bounds for various tuning methods and present several examples. This framework can be utilized as a theoretical justification for special-designed methods in subsequent works.
>
> **6. Why not choose other methods?**
>
> Thank you for the insightful question. As Sec. 5, we mitigate the tuning bias by reducing the number of parameters. A specific example is to use alternative methods with fewer parameters (e.g., switching from VS to TS). However, this way may be impractical when complex methods are necessary to obtain tighter CP sets (ConfTr), improved model calibration (VS), and so on. This question highlights our contribution to revealing the scaling law of tuning bias, which provides guidelines for designing the tuning process in various scenarios.
>
> **7. Why not split the dataset?**
>
> Thank you for raising the concern. We refer to reviewer jecM's response #3 to answer the issue. As suggested by the reviewer, we will strengthen the motivation in Sec. 2 of the final version.
>
> [1] Villalobos, Pablo. "Scaling Laws Literature Review." Published online at Epochai. org (2023).

---

> > ### Comment · Reviewer_jaH5 · 2025-04-02
> >
> > Thanks for the authors' responses, and most of my concerns have been addressed. I agree with  Reviewer 4yEA for the contribution of this work, willing to raise my original score to 4.

---

> > > ### Author Response · Authors · 2025-04-02
> > >
> > > Thank you for reviewing our response and increasing the score. We are delighted that our response addressed your concerns. Your feedback is highly valuable in improving the quality of this work.

---

### Official Review · Reviewer_4yEA · 2025-03-13

**Overall Recommendation:** 5

**Summary:**

In this paper, the authors focus on the tuning bias produced by parameter tuning in many conformal preidction methods. First, they  reveal that the tuning bias is negligible for simple parameter tuning in many conformal prediction methods. Then, the authors establish a parametric scaling law, showing that tuning bias increases with parameter space complexity and decreases with calibration set size, supported by both empirical evidence and a theoretical framework using constrained ERM and VC dimension. The paper also discusses solutions to mitigate tuning bias, such as increasing the calibration set size or reducing parameter space complexity through fewer parameters or regularization techniques.

**Claims And Evidence:**

The claims made in the submission are generally supported by clear and convincing evidence, including a combination of empirical results, theoretical analysis, and practical considerations.

**Essential References Not Discussed:**

No

**Experimental Designs Or Analyses:**

The experimental designs and analyses in the paper are generally sound and valid. This work can be benefited if the authors can provide more detailed information on hyperparameter settings, such as the number of repetitions. I believe it would improve the reproducibility and robustness of the findings.

**Methods And Evaluation Criteria:**

The evaluation criteria make sense for the problem of tuning bias in conformal prediction. The paper conducts extensive empirical studies across various conformal prediction methods (e.g., RAPS, SAPS, score aggregation, temperature scaling, vector scaling, C-Adapter, ConfTr) on benchmark datasets such as CIFAR-10, CIFAR-100, and ImageNet. These datasets are standard in machine learning and are suitable for evaluating the performance of conformal prediction methods. By varying the calibration set size and the complexity of the parameter space, the paper effectively demonstrates the impact of these factors on tuning bias, which aligns with the theoretical analysis.

**Other Comments Or Suggestions:**

N/A

**Other Strengths And Weaknesses:**

Strengths
1. The problem studied in this paper is significant. It is practical to use the same hold-out data for conformal calibration and parameter tuning in data-scarce scenerios. Therefore, the findings have practical implications for improving the reliability of conformal prediction in real-world applications.
2. The theoretical analysis is solid. The paper’s theoretical framework offers a new perspective on tuning bias, which could influence future research in statistical learning and uncertainty quantification.
3. The paper is well-structured, with clear delineation between empirical studies, theoretical analysis, and practical solutions
4. The proposed guideline is useful. The discussion of potential solutions to mitigate tuning bias, such as increasing the calibration set size and reducing parameter space complexity through regularization, provides actionable insights for practitioners.


Weaknesses
1. (Minor issue) The writing of experimental results is a little ambiguous. I encourage the authors to improve the writing of Subsections 3.1 and 3.2, providing more details of the experimental settings and clearer observations.

**Questions For Authors:**

Please explain why C-adapter achieves much smaller tuning bias but vector scaling cannot? It seems C-adapter tunes more parameters than VS.

**Relation To Broader Scientific Literature:**

In summary, the paper makes several key contributions that advance the understanding of tuning bias in conformal prediction, both theoretically and empirically. In particular, this work provides new insights and practical guidelines for managing tuning bias in real-world applications:
1. Analyzing the Tuning Bias in Conformal Prediction;
2. Presenting Parametric Scaling Law of Tuning Bias;
3. Theoretical Framework to quantify the upper bound;
4. Practical Guidelines to reduce tuning bias.

Although some works investigated the non-exchangeability of using the same dataset(as stated in the Introduction), this work is novel as it provides a extensive study on quantifying the tuning bias of conformal prediction, which could be pretty interesting in the community.

**Theoretical Claims:**

The theoretical claims are supported by correct and logically sound proofs, grounded in established statistical learning theory. For example, the derivation of the scaling law is supported by Proposition 4.2 and Proposition 4.6, which provide bounds for the tuning bias in finite and infinite parameter spaces, respectively. These propositions use classical concentration inequalities and VC dimension bounds. While the proofs could benefit from more detail and rigor in certain steps, the overall correctness of the theoretical analysis is robust.

---

> ### Author Rebuttal · Authors · 2025-03-31
>
> We thank the reviewer for the positive feedback. Below, we address your concerns point by point.
>
> **1. C-Adapter vs. Vector Scaling**
>
> > Please explain why C-adapter achieves much smaller tuning bias but vector scaling cannot? It seems C-adapter tunes more parameters than VS.
>
>
> Thank you for the insightful question. It prompted us to delve deeper into the structural differences between C-Adapter and Vector Scaling (VS). In particular, we find that the **order-preserving regularization** in C-Adapter can significantly decrease the tuning bias. Formally, we propose a new proposition:
>
> **Proposition:** *Let $f$ be a logits value function for a classification of $K$ classes. The matrix scaling $g(x) = W f(x) + b$ is order-preserving if and only if $W$ has the form $W = a I + \mathbf{1} v^T$ for some scalar $a > 0$ and vector $v \in \mathbb{R}^K$, and $b$ is a constant vector (i.e., $b_j = b_{j'}$ for all $j, j'\in [K]$). Here, $I$ is the $K \times K$ identity matrix and $\mathbf{1}$ is the $K$-dimensional vector of all ones.*
>
> Here, we regard C-Adapter as a special case of matrix scaling with order-preserving regularization. The above proposition shows that **the order-preserving regularization reduces the dimension of parameter space from $K^2 +K$ to $K+2$**, which is much smaller than VS  with its dimension being $2K$. Based on the parametric scaling law (Section 3), we explain why C-Adapter can achieve lower tuning bias than VS. Thank you again for inspiring us to revealing the impact of order-preserving regularization and we will add it in the discussion of the final version.
>
> **2. Experimental details & reproducibility**
>
> > The experimental designs and analyses ... can be benefited if the authors can provide more detailed information on hyperparameter settings, such as the number of repetitions. I believe it would improve the reproducibility and robustness of the findings.
>
> Thank you for the suggestion. We provide the experimental settings in Appendix A. In particular, we repeat all experiments with 30 runs and present the standard deviations in Fig. 2. In the final version, we will improve the writing of experimental setup to enhance the clarity and reproducibility.
>
> **3. Writing clarity (subsections 3.1/3.2)**
> > (Minor issue) The writing of experimental results is a little ambiguous. I encourage the authors to improve the writing of Subsections 3.1 and 3.2, providing more details of the experimental settings and clearer observations.
>
> Thank you for pointing out the writing issue. We will revise Subsections 3.1 and 3.2 to clearly present the experimental settings and the key observations, in the final version.

---

> > ### Comment · Reviewer_4yEA · 2025-04-02
> >
> > I thank the authors for the detailed responses. My concerns are clarified now. Given the contribution of this excellent work, I support the acceptance now.

---

> > > ### Author Response · Authors · 2025-04-02
> > >
> > > Thank you for reviewing our response and raising your score. We are pleased that our response addressed your concerns, which also improves the quality of this work.

---

### Official Review · Reviewer_mAZq · 2025-03-16

**Overall Recommendation:** 3

**Summary:**

This paper points out the problem that the exchangeability assumption of conformal prediction does not hold if the holdout set (applied for parameter tuning) and calibration set are identical. A parametric scaling law is proposed such that the tunning bias increases with parameter space complexity and decreases with calibration set size. A theoretical study is conducted to provide an upper bound of the bias. Potential solutions, like regularization during tunning, are provided.

**Claims And Evidence:**

Claims made in the submission are supported by clear and convincing evidence.

**Essential References Not Discussed:**

The related works are introduced extensively.

**Experimental Designs Or Analyses:**

Experiments are only conducted on two datasets CIFAR-100 and ImageNet, and there is no regression task. It should be claimed if the work only focuses on conformal prediction on classification tasks. There is no standard deviation results in Table 1. The authors do not mention the experiments in Table 1 were conducted multiple times.

**Methods And Evaluation Criteria:**

This work does not propose an implementable solution (i.e. an algorithm). Only an intuitive solution is provided.

The notation of a prediction set is not consistent. It shows up \hat{C} and C in different places. Do they refer to different concepts?

The metric CovGap is introduced clearly. In line 80, as prediction set C(x) is related to the test input x. However, CovGap(C) is only a function of C and x disappears. Do you mean CovGap is the expectation of coverage gap for all test inputs? The same problem happens to other metric definitions, including TuningBias(C).

The empirical study is conducted on APS score, but the score is not cited in line 163 until Appendix A.

**Other Comments Or Suggestions:**

There is a typo in line 246. The notation system should be redesigned. Also, the name of corollary and proposition is abused. Intermediate conclusions or experiment observations should not be stated as corollaries and propositions (such as proposition 5.1 and 5.2).

**Other Strengths And Weaknesses:**

The investigated problem is novel and sufficient theoretical works are conducted.

However, the presentation should be further improved. For instance, the same notations are reused for different concepts, such as \hat{C} in Eq.(3) and Eq.(5).  Legends for 'same' and 'holdout' are not consistent in figure 2 and 3. Writing should be improved. In line 367, 'Theoretically. we provide a theoretical result...'. The poor presentation makes the paper hard to understand. It hinders me to validate the correctness of some theories and proofs.

Besides, as mentioned above, the works lacks a more insightful solution to the problem. The experiments of the current intuitive solutions are not enough either.

**Questions For Authors:**

What space of \mathbb{E}\mathcal{R}_\Lambda is computed over?
CovGap is a function of alpha in line 233. Does CovGap is also a function of alpha in line 81 as well?

**Relation To Broader Scientific Literature:**

The problem of overfitting the holdout set, which should not be applied for calibration, is kind of novel, and I appreciate the experimental and theoretical works. The goal of reducing the amount of parameters and regularization is to prevent the overfitting issue. People can easily come up with the ideas without the theoretical analysis. More sophisticated solutions are expected for higher impact.

**Theoretical Claims:**

Based on Eq.(2), a ceiling function should be added at the subscript of Q in Eq.(4).
The upper bound \mathbb{E}\mathcal{R}_\Lambda is not introduced clearly when it shows up first in line 232 until line 260. Also, it is not clarified what space/set the expectation is calculated on.

---

> ### Author Rebuttal · Authors · 2025-03-31
>
> We thank the reviewer for the nuanced and constructive feedback. Below, we address your concerns point by point.
>
> **1. Lack of a sophisticated solution**
>
> We want to clarify that the primary objective of this work is to provide a comprehensive understanding of tuning bias in conformal prediction rather than to develop a specific, sophisticated solution at this stage. To this end, we present the main contributions of this study as follows: identifying the tuning bias, introducing a scaling law for the tuning bias, and establishing a theoretical framework to quantify its upper bound. Furthermore, we discuss potential solutions for mitigating tuning bias as an extension. Rather than introducing a novel methodology, we propose **two effective guidelines** to address tuning bias in real-world applications: reducing the parameters of model tuning (e.g., adopting a more parameter-efficient strategy) and implementing regularization techniques (e.g., order-preserving constraint). These guidelines provide **actionable**, **theory-driven** solutions and establish a foundation for future research to develop specialized methods tackling this challenge. Therefore, we believe this work not only **builds a robust theoretical framework** for understanding tuning bias—as recognized by Reviewer jaH5—but also **charts a clear path** for subsequent studies in this field.
>
> **2. Task scope**
>
> Thank you for the suggestion. We clarify that our analysis reveals a general phenomenon of conformal prediction in both classification and regression tasks. Section 3 focuses on classification, as many parameter tuning methods (like TS/VS and ConfTr) are designed for classification tasks. Here, we provide new results to show the case of regression tasks following previous work [1] with a target coverage of 90% and 30 repetitions. We present the CovGap and TuningBias (in percentage) in the table below:
>
> |Method|Varying #models||Varying Cal. Size||
> |-|-|-|-|-|
> ||40|160|100|500|
> |Same|5.63|6.27|5.41|3.27|
> |Hold-out|3.17|2.56|3.91|2.90|
> |**Tuning-bias**|2.46|3.71|1.50|0.37|
>
> From the table above, we validate the parametric scaling law of tuning bias on regression tasks. In addition, our theoretical framework is general for various tasks. We will clarify the task scope and add the above results and details of the experiments in the final version.
>
> **3. Table 1 - Standard deviations/repeats**
>
> Thanks for the suggestion. In the current version, we present average results with few runs in Table 1. Here, we updated the results (in percentage) with 30 runs:
>
> | Method | CIFAR-100 | ImageNet |
> |-|-|-|
> |TS| **0.59 ± 0.38** | **0.43 ± 0.29** |
> |VS| 1.63 ± 0.76 | 6.43 ± 0.53 |
> | ConfTr (ft.) w/ OP | **0.52 ± 0.37** | **0.40 ± 0.31** |
> | ConfTr (ft.) w/o OP | 6.15 ± 0.86 | 21.68 ± 0.58 |
>
> The new results lead to the same conclusion as the previous version. We will update the table in the final version.
>
> **4. Performance gap between various datasets**
>
> Thank you for the suggestion. It is wothy noting that the parameter numbers of those tuning methods (e.g., VS and C-Adapter) are positively related to the class numbers of the dataset (See Line 354). Thus, datasets with more classes require more parameter numbers in the tuning, leading to a larger tuning bias. This explains why those methods perform differently in various datasets. We will add a brief explanation in Appendix B of final version.
>
> **5. Concerns of presentation**
>
> Thank you for raising the writing concerns. We agree that a clearer and more consistent notation system can benefit the paper a lot:
> 1. **CovGap**: Both CovGap and TuningBias are dataset(distribution)-level metrics, instead of instance-level. In particular, CovGap measures the absolute difference between the target coverage and the actual coverage on the dataset/distribution, see Line 81. In addition, CovGap is a function of alpha, but we omit the alpha for simplicity. We will fix the notation issue in Lines 80-81 and update the notation to ensure consistency in the final version.
> 2. **Prediction set**: $\hat{C}$ in Eq. (3) and (5) are the empirical form of the CP set, where the *hat* notation emphasizes it is associated with observations. $C$ denotes general form CP sets that may be assigned without observations (e.g., oracle CP sets).
> 3. **Clarify $\mathcal{R}_\Lambda$**: It is the supremum of an empirical process. In the current version, we define it in Line 239 and rewrite it in Line 260.  The  $E\mathcal{R}_\Lambda$ is expected over the distribution of the calibration set.
> 4. **Propositions**: Prop. 5.1 and 5.2 are theoretical derivations (See proofs in App. I and J) instead of empirical observations.
> 5. Other issues: we will add proper citations, fix the typos, and improve the clarity in the final version. Thank you again for the nuanced review.
>
> [1] Liang, Ruiting, Wanrong Zhu, and Rina Foygel Barber. “Conformal Prediction after Efficiency-Oriented Model Selection.” arXiv. https://doi.org/10.48550/arXiv.2408.07066. (2024)

---

### Official Review · Reviewer_jecM · 2025-03-17

**Overall Recommendation:** 4

**Summary:**

This paper finds out that the coverage gap of using same dataset for tuning and calibration is negligible in most of the conformal prediction methods. Also, this paper observes a scaling law about how parameter space complexity and calibration set size influence the tuning bias. Then this paper proposes a theorectical framework to quantify tuning bias, and gives out a theoretical proof for the scaling law. At the end, this paper discuss two solutions of reducing tuning bias based on the scaling law.

**Claims And Evidence:**

The claims are clear, and the evidence is convincing.

**Essential References Not Discussed:**

All the essential references are discussed.

**Experimental Designs Or Analyses:**

Although the paper discusses how tuning bias scales with the complexity of the parameter space, there might be a lack of experiemental results that can show how reducing the number of parameter could help reducing the bias.

**Methods And Evaluation Criteria:**

For the methods, there is a risk of overfitting to the specific characteristics of the calibration data used in the study, particularly if the separation between tuning and calibration is not well-managed. This could lead to models that perform well on specific dataset characteristics but poorly generalize to new data, undermining the reliability and utility of the predictions in practical applications. These drawbacks underscore the need for careful application and further testing of these methods across various settings and conditions to fully understand their limitations and potential.

**Other Comments Or Suggestions:**

There are not too much typos or notation error, but I recommend to make the TuningBias and CovGap in the formula as Text form.

**Other Strengths And Weaknesses:**

I think this paper is not well-motivated, the paper may not sufficiently demonstrate the practical necessity of investigating tuning bias where the tuning and calibration datasets are the same. In most real-world applications and existing research, these datasets are intentionally kept separate to avoid overfitting and ensure the model's generalizability. The necessity to study what happens when they overlap might not be convincingly argued, making the motivation behind this research seem theoretical rather than practical. If the paper leans heavily on theoretical justifications without clear paths to application or examples of real-world scenarios where such tuning bias issues prominently occur, it might reinforce the impression that the motivation is more academic than practical.

**Questions For Authors:**

No questions.

**Relation To Broader Scientific Literature:**

While the paper contributes interesting findings on tuning bias and its scaling laws, it may not adequately integrate these contributions with existing theories or frameworks within the broader field of machine learning. Even in the field of conformal prediction, how the findings help the further research path is not mentioned.

**Theoretical Claims:**

The theoretical claims are solid with proofs.

---

> ### Author Rebuttal · Authors · 2025-03-31
>
> Thanks for your positive and valuable feedback.
>
> **1. Results of reducing parameter numbers:**
>
> Thank you for the suggestion. In the manuscript, we presented two empirical evidences to validate the effect of reducing the parameter number:
>
> 1. TS vs. VS (Table 1, Fig. 1 and 3): we present a pilot study to show that TS with fewer parameters achieves much smaller tuning biases than VS with more parameters.
> 2. Experiments of scaling law (Fig. 2(a)): we analyzed the correlation between amounts of parameters and tuning bias by freezing different numbers of parameters within VS. The results show that increasing the number of parameters leads to higher tuning bias, supporting the claim.
>
> As the analysis above is sufficient to show that "reducing the parameter numbers can result in lower tuning bias", we did not present more results in Section 5. In the final version, we will explicitly refer to these empirical results in the discussion.
>
> **2. Relation to literature:**
>
> Thank you for the positive comment and suggestion. In the final version, we will improve the writing of related work to clearly present the position of this work in the literature. Here, we contextualize our work as follows:
>
> - **Integration with existing theories**: we formulated the tuning bias as *constrained ERM problem*, a special case of learnability theory (we discussed in related work). In Subsec. 4.2 and 4.3, we introduce *Dvoretzky–Kiefer–Wolfowitz inequality* and *VC dimension* to analyze the tuning bias in finite and infinite parameter spaces, respectively. Thus, our theoretical framework is tightly integrated with existing machine learning theories, as discussed in related work.
> - **How it benefits future works**: In this work, we provide the first study to quantify the tuning bias and its scaling laws. This enables researchers to determine when splitting the dataset is necessary or if data reuse is acceptable, which is particularly crucial in data-scarce scenarios (such as rare diseases in medical diagnosis). In addition, we also provide practical guidelines for developers to alleviate the tuning bias (as appreciated by Reviewer 4yEA). Theoretically, this work is the first to employ the ERM framework in conformal prediction, offering a novel tool for analyzing the learnability of general conformal prediction problems.
>
> **3. The practical necessity of exploring tuning bias and why not split dataset**
>
> Thank you for highlighting this writing issue. In the current version, we only describe the significance of reducing tuning bias in the discussion (Sec. 5), which makes it challenging for readers to grasp the motivation earlier in the paper. Here, we'd like to clarify that understanding the tuning bias is crucial in conformal prediction practice:
>
> - **Data-scarce scenarios**: Splitting the labeled dataset is impractical in data-scarce scenarios like rare diseases, natural disaster prediction, and privacy-constrained personal data. With limited data, using separated datasets will reduce points for parameter tuning and conformal calibration, compromising the approach's effectiveness and stability. Thus, it’s valuable to assess when splitting is needed, or data reuse is permissible rather than sticking to traditional practices.
> - **Simple implementation**: Even with sufficient data, maintaining separate sets can increase the pipeline complexity. Understanding when this separation is unnecessary—such as when tuning bias is negligible—enables simpler, more streamlined workflows while preserving coverage guarantees, offering practical relevance.
> - **Foundational understanding**: exploring the tuning bias can provide an in-depthknowledgeg of the exchangeability assumption in conformal prediction. In particular, the insight in this work may inspire future works in non-exchangeable conformal prediction.
>
> It could also answer Reviewer jaH5's concern, "why not split split the dataset?" for the same reasons.  Indeed, it is easy to split the validation set when the provided data is sufficient. However, it can be particularly important to consider data reusing in data-scarce scenarios, like rare diseases, natural disaster prediction, and privacy-constrained personal data. With limited data, separating the dataset will further exacerbate the data scarcity problem, compromising the effectiveness of both conformal calibration and parameter tuning. In addition, exploring the tuning bias can provide an in-depth understanding of the exchangeability assumption, which may inspire future works in non-exchangeable CP.
>
> As suggested by the reviewer, we will emphasize the motivation of exploring the tuning bias in the Introduction and Background of the final version.
>
> **4. Supplementary material and notation**
>
> Thanks for the suggestion. In the final version, we will update the notation of TuningBias and CovGap with text form and release the code on GitHub once it is accepted.

---

> > ### Comment · Reviewer_jecM · 2025-04-03
> >
> > Thanks for authors' response, I can understand the motivation now. Hope this can be highlighted in the final version, and I will adjust my score.

---

> > > ### Author Response · Authors · 2025-04-03
> > >
> > > Thank you for raising the score. We are pleased that our response addressed your concerns, which also improves the quality of this work. Once again, we appreciate your positive and valuable feedback.

---

### Decision · Program_Chairs · 2025-05-01

**Decision:**

Accept (poster)

**Comment:**

This paper tracks an important problem: conformal prediction which is a popular framework of uncertainty quantification that constructs prediction sets with coverage guarantees. The authors establish a theoretical framework to quantify the tuning bias and provide rigorous proof for the scaling law of the tuning bias by deriving its upper bound.

In the initial reviews, reviewers have some concerns regarding the presentation of this paper and request some clarifications. After the rebuttal, all reviewers agree that this paper should be accepted due to its novelty.

The authors are encouraged to merge all comments to further improve the presentation quality of this paper. I nominate this paper for oral presentation due to the significance of the researched topic.